# Multiscale modelling of chromatin 4D organization in SARS-CoV-2 infected cells

Andrea M. Chiariello[1,4] ✉, Alex Abraham [1,4], Simona Bianco [1], Andrea Esposito [1], Andrea Fontana [1], Francesca Vercellone [2], Mattia Conte[1] & Mario Nicodemi[1,3] ✉

SARS-CoV-2 can re-structure chromatin organization and alter the epigenomic landscape of the host genome, but the mechanisms that produce such changes remain unclear. Here, we use polymer physics to investigate how the chromatin of the host genome is re-organized upon infection with SARS-CoV-2. We show that re-structuring of A/B compartments can be explained by a re-modulation of intra-compartment homo-typic affinities, which leads to the weakening of A-A interactions and the enhancement of A-B mixing. At the TAD level, re-arrangements are physically described by a reduction in the loop extrusion activity coupled with an alteration of chromatin phase-separation properties, resulting in more intermingling between different TADs and a spread in space of the TADs themselves. In addition, the architecture of loci relevant to the antiviral interferon response, such as *DDX58* or *IFIT*, becomes more variable within the 3D single-molecule population of the infected model, suggesting that viral infection leads to a loss of chromatin structural specificity. Analysing the time trajectories of pairwise gene-enhancer and higher-order contacts reveals that this variability derives from increased fluctuations in the chromatin dynamics of infected cells. This suggests that SARS-CoV-2 alters gene regulation by impacting the stability of the contact network in time.

The SARS-CoV-2 outbreak had an important impact on society and science. Several efforts have been made to understand the effects of the virus on host cells from different points of view, ranging from studying the immunological response to the virus[1] to investigating the effects of infection on epigenetic regulation[2] or researching therapeutic molecular targets[3]. SARS-CoV-2 is able to impact the chromatin architecture[4,5] of the host cell, which in general is an important control layer for gene regulation[6,7]. Indeed, virus infection has been shown, for instance, to alter genome organization of olfactory receptors in humans and hamsters, providing a potential mechanism to explain anosmia[5], one of the typical symptoms of Covid-19. More recently, it has been shown that

SARS-CoV-2 deeply impacts genome organization at multiple length scales, ranging from some kilobases up to A/B compartment level, and influences the activity of gene categories[4] fundamental to the immunological response[1], such as genes involved in the interferon (IFN) response and pro-inflammatory genes. Although those studies shed light on the effects of SARS-CoV-2 on genome organization, the physical mechanisms regulating how the virus changes the host cell 3D chromatin structure are not clearly understood.

Here, we employ models from polymer physics[8,9] and Molecular Dynamics (MD) simulations to quantitatively study multiscale chromatin re-arrangements resulting from SARS-CoV-2 infection of the

[1]Dipartimento di Fisica, Università degli Studi di Napoli Federico II, and INFN Napoli, Complesso Universitario di Monte Sant'Angelo, 80126 Naples, Italy. [2]Dipartimento di Ingegneria Elettrica e delle Tecnologie dell'Informazione - DIETI, Università degli Studi di Napoli Federico II, and INFN Napoli, Via Claudio 21, 80125 Naples, Italy. [3]Berlin Institute for Medical Systems Biology at the Max Delbruck Center for Molecular Medicine in the Helmholtz Association, Berlin, Germany. [4]These authors contributed equally: Andrea M. Chiariello, Alex Abraham. ✉e-mail: andreamaria.chiariello@na.infn.it; mario.nicodemi@na.infn.it

host cell. In general, polymer models have shown to be a valuable tool to investigate genome organization in the cell nucleus, as they are able to describe the physical mechanisms shaping chromosome folding[9] and to explain several features of genome architecture, e.g., the heterogeneity of chromatin structure in single cells[10] or the structural rearrangements caused by genomic mutations and their impact on gene expression[11]. At very large genomic length scales (several Mbs), we show that a simple polymer made of consecutive compartments (i.e., block-copolymer model[12,13]), in which homo- and hetero-typic interactions are defined within and between compartments, is able to explain the weakening of A compartment and enhancement of A-B mixing[4] experimentally observed in SARS-CoV-2 infected genome, by basically reducing the intra-compartment homo-typic A-A affinity. At TAD level (from hundreds Kbs to some Mbs), we show that a model combining loop-extrusion[14,15] and phase-separation[13,16] effectively describes the experimentally observed intra-TAD weakening in SARS-CoV-2 infected cells[4], which results from a reduction of extruders density coupled with an alteration of phase-separation properties of chromatin filament. Importantly, we find that this alteration is not observed in a polymer model describing chromatin organization in human coronavirus HCoV-OC43 infected cells (causing common cold), suggesting that alteration of phase-separation is a peculiar feature of SARS-CoV-2 infection. Furthermore, using the same model informed with HiC data[10,11], we investigate the architecture of genomic loci containing *DDX58* and *IFIT* genes, which are of relevant immunological interest since linked to the antiviral interferon (IFN) response[17] of the host cell. Specifically, analysis of polymer structures reveals that in SARS-CoV-2 model the population of single-molecule 3D configurations results more variable and less coherent with respect to non-infected condition, suggesting that the alteration of activity observed for IFN genes[1,4] can be due to a general loss of structural specificity caused by alteration of physical mechanisms driving 3D chromatin

organization. By leveraging on our Molecular Dynamics simulations, we show that the model of SARS-CoV-2 exhibits a more scattered time dynamics, leading to a reduction of contact stability between pairs or hubs of multiple regulatory elements.

Overall, our polymer-physics based study provides insights into how viral infection affects chromatin organization and suggests that this occurs through the combined alteration of the loop-extrusion and phase-separation properties of chromatin, indicating a potential mechanistic link between the observed genome re-structuring and mis-regulation of, e.g., key genes involved in the immunological response within the host cell.

## Results

We study chromatin re-organization of host cell genome infected by SARS-CoV-2. To this aim, we consider recently published HiC data[4] in control condition, i.e., not infected human A549 cells expressing ACE2 (referred to as Mock) and in human A549 cells expressing ACE2 at 24-h post SARS-CoV-2 infection, in which HiC data highlighted re-arrangements at multiple length scales, involving A/B compartment, TADs and regulatory contacts within specific loci[4].

### Modeling of chromatin re-structuring in A/B compartments

One of the main structural re-arrangements on chromatin architecture resulting from SARS-CoV-2 infection of the host genome occurs at A/B compartment level. Specifically, it has been observed that viral infection results in a general weakening of A-compartment concomitantly with an enhanced A/B compartment mixing[4], as schematically depicted in Fig. 1a. To quantitatively investigate such effect, we first focused on a simple model of chromatin at A/B compartment level. We employed the Strings and Binders Switch polymer model[13,18], where chromatin folding is driven by a phase-separation mechanism (Methods), similar to other models proposed for chromatin

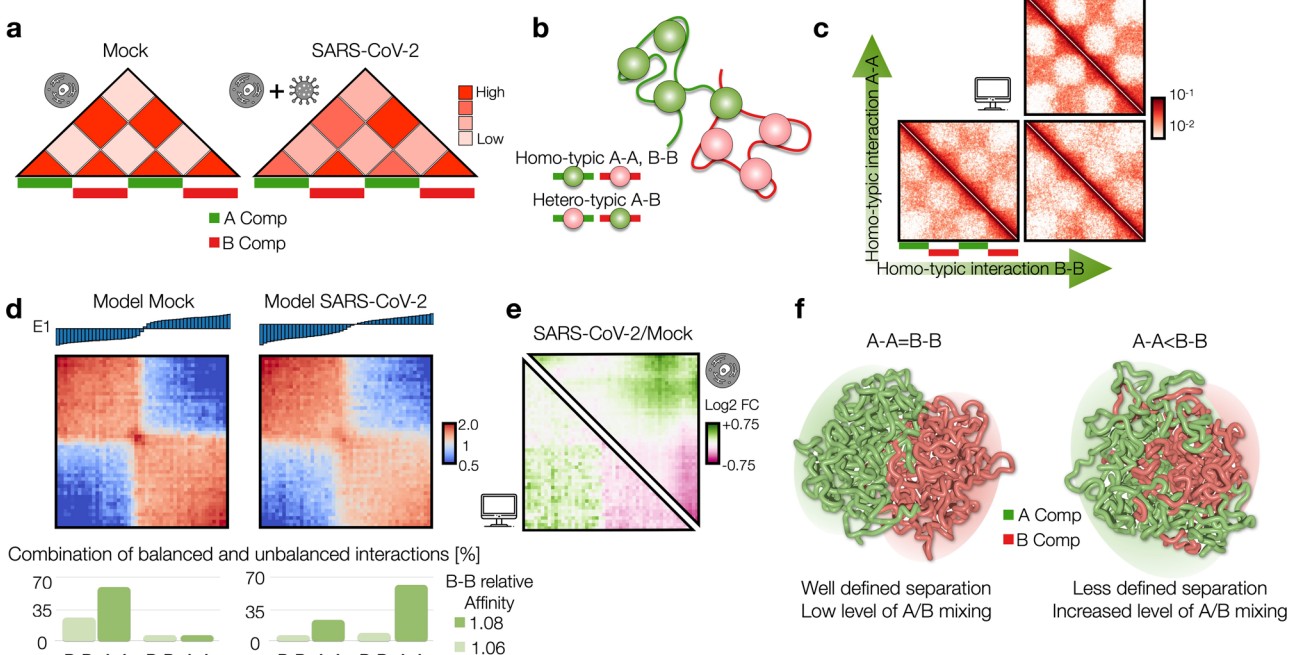

**Fig. 1 | Modeling of chromatin re-structuring in A/B compartments. a** A/B compartment mixing in SARS-CoV-2 infected genome detected from HiC data[4]. **b** Polymer model of A/B compartment envisages homo-typic (A-A and B-B) and hetero-typic (A-B) interactions. **c** Variation of homo-typic binding affinity results in weakening of A-compartment and enhancement of A/B compartment mixing. Heatmaps are computed from a populations of 3D structures obtained from MD simulations. **d** Saddle-plot of best fit polymer model for Mock and SARS-CoV-2

conditions. Above, the sorted 1st eigenvector E1 is shown. Below, best fit coefficients obtained to fit experimental saddle-plots. Dark and light green bars indicate B-B affinity used in the fit, normalized with respect the background hetero-typic interaction (Methods). **e** Log2 Fold Change of the saddle-plots from the model (bottom left) and HiC data[4] (top right). Pearson correlation between matrices is $r = 0.77$. **f** 3D rendering of best polymer models for Mock (left) and SARS-CoV-2 (right). Source data are provided as a Source Data file.

compartmentalization[12,19–21]. Briefly, we consider a simple block copolymer where A and B compartments are modeled as two different types of binding sites (represented as different colors) which can homo-typically interact with cognate molecules (named binders) with an affinity $E_{A-A}$ and $E_{B-B}$, driving A-A or B-B interactions within the same compartment (Fig. 1b). On the other hand, binders can also mediate A-B or B-A hetero-typic interactions, with a general affinity $E_{A-B}$ (Methods). To ensure micro-phase separation of A and B blocks, we always consider $E_{A-A} > E_{A-B}$ and $E_{B-B} > E_{A-B}$[19]. We first considered models with balanced interactions $E_{A-A} = E_{B-B}$ and varied the homo-typic affinity (here, hetero-typic affinity $E_{A-B}$ is kept constant, Methods). In general, low homo-typic interactions result in a reduced compartmentalization and increased A/B mixing, as shown by the model contact maps (Fig. 1c and Supplementary Fig. 1a), the first eigenvector E1 from Principal Component Analysis (PCA) and the saddle-plots of the sorted eigenvector components[22,23] (Supplementary Fig. 1b, Methods). Analogous effects are observed by increasing hetero-typic affinity $E_{A-B}$, keeping constant homo-typic $E_{A-A} = E_{B-B}$ (Supplementary Fig. 2, Methods). In addition, models with unbalanced interactions with $E_{B-B} > E_{A-A}$ result in both A/B mixing and, importantly, in weakening of A-compartment shown in the contact maps (Fig. 1c, Supplementary Fig. 3a) and in asymmetric saddle plots (Supplementary Fig. 3b). Therefore, we reasoned that a combination of models with balanced and unbalanced interactions can fit A/B compartment alteration in SARS-CoV-2 infected genomes. We then fitted the best combination of interactions to reproduce the average compartment profile (using saddle-plot maps) obtained from HiC data in Mock and SARS-CoV-2-infected cells (Methods). Interestingly, Mock HiC data are mainly described (almost 90%) by a model with balanced homo-typic interactions (i.e., $E_{A-A} = E_{B-B}$ Fig. 1d, bottom left panel), indicating a similarity in the A and B average compartmentalization level and consistent with existing models of A/B compartmentalization[20]. Conversely, data in SARS-CoV-2 infected cells are best described by a combination of unbalanced homo-typic interactions where $E_{B-B} > E_{A-A}$ ( > 60%) consistently with the general weakening of A-compartment and above-mentioned enhanced A/B mixing, with balanced interactions only marginally involved (about 20%, Fig. 1d, bottom right). Importantly, albeit very simple, this model exhibits a high level of agreement with experimental data, as shown by the comparison between Log2 FC (SARS-CoV-2/Mock) of saddle-plot matrices (Pearson $r = 0.77$, Fig. 1e). Analogous results are found by fitting a combination of different hetero-typic affinities, keeping fixed balanced interactions $E_{A-A} = E_{B-B}$. Indeed, we find that SARS-CoV-2 data are better described by a combination with higher hetero-typic affinities with respect to the Mock case (Supplementary Fig. 4a), although the saddle-plot changes are captured with less accuracy (Pearson $r = 0.6$, Supplementary Fig. 4b), indicating an important role for the model with unbalanced affinities.

To test the robustness of our results on a real genomic region, we repeated the above discussed analysis using as case of study chromosome 11, with A and B blocks defined using the 1st eigenvector from PCA (Supplementary Fig. 5a, Methods). MD simulations of this model return contact maps accurately describing A/B compartment profile contained in the HiC data (Supplementary Fig. 5b, c, Methods). Specifically, we find that Mock data are best described by a combination with ~70% balanced interactions (Supplementary Fig. 5d), in line with the previously discussed result but also highlighting a not negligible role for unbalanced interactions even in the not-infected case, likely due to the distinct machineries behind A and B compartments formation[24]. Conversely, SARS-CoV-2 data are best described by a combination with higher level of unbalanced affinities (~80%, Supplementary Fig. 5d, e) in agreement with the weakening of A-compartment. Overall, these results show that chromatin re-arrangements observed in infected host genome can be explained by a re-modulation of affinities which in turn affects the tendency of compartments to microphase separate, as also shown by the 3D

rendering of polymer structures representing A and B compartments in Mock (Fig. 1f, Supplementary Figs. 4d, 5f, left panel) and SARS-CoV-2 (Fig. 1f, Supplementary Figs. 4d, 5f, right panel) infected conditions.

## Viral infection impacts loop-extrusion and phase-separation features at TAD level

Next, we investigated how SARS-CoV-2 infection impacts genome organization at TAD level, i.e., genomic scales ranging from tens of kbs to some Mbs. Indeed, it has been shown that viral infection produces a general weakening of intra-TAD contacts along with a slightly increase of inter-TADs interactions[4] (Fig. 2a) and concomitantly with a general reduction of Cohesin level[4], suggesting a reduction of loop-extrusion activity. To test this hypothesis and give a mechanistic insight to this result, we used a polymer physics model combining both loop-extrusion[14,15] (LE) and phase-separation[13,18] (PS) mechanisms (Fig. 2b, Methods), which recently has been shown to successfully describe chromatin organization at single cell level[10]. In this scenario, LE and PS simultaneously act and the pattern of chromatin contacts observed in HiC data results from an interplay between both processes (Fig. 2c). By varying the main system parameters, i.e., interaction affinity and average distance between extruders (or equivalently their number, Methods) (Supplementary Fig. 6a), we generated several different polymer populations with their simulated contact maps (Supplementary Fig. 6b) and contact probability profiles (Supplementary Fig. 6c). In this way, we were able to identify the polymer model best fitting the contact probability obtained from HiC data (Methods), in the genomic distance ranging from the sub-TAD level (approx. 10 kb) to inter-TADs contacts (some Mbs, Methods). The model is able to explain with accuracy experimental data, as shown by the fit of the average contact probability (as shown by $\chi^2$ values in Supplementary Fig. 6d) in Mock (Fig. 2d, left bottom panel) and in SARS-CoV-2 infected (Fig. 2d, right bottom panel) conditions. Importantly, the best model describing Mock data revealed an average distance between extruders of approximately 100–150 kb (Supplementary Fig. 6d, left panel), consistent with previous estimates obtained from other HiC datasets[15]. Conversely, the best model fitting the SARS-CoV-2 infected HiC data was best described by a consistently decreased number of extruders (approximately halved, Supplementary Fig. 6d, right panel), in full agreement with experimental observations where viral infection produces a genome-wide decrease of Cohesin levels[4]. Interestingly, this analysis revealed that, in order to fit HiC data in infected cells, the reduction of extruders is coupled with a reduction of interaction affinity (around 15–20%) between binders and chromatin (Supplementary Fig. 6d), which affects chromatin spatial localization and contributes to the general weakening of intra-TADs contacts (Fig. 2d, upper panels) observed in infected genomes. This is quantitatively shown in the Log2 FC (SARS-CoV-2/Mock) of contact maps (Fig. 2e, upper panel) and contact probabilities (Fig. 2e, bottom panel), which exhibits a very good agreement with experimental data (Pearson $r = 0.82$, Methods). To check whether such changes in chromatin architecture are peculiar of SARS-CoV-2 or if they are observed in other coronaviruses, we repeated the above-described analysis using HiC data in cells infected by the human coronavirus (HCoV) OC43[4], which causes common cold. Again, we were able to fit the average contact probability with high accuracy (chi-square test p-val=1). Intriguingly, we find that the best model describing HCoV-OC43 infection is analogous to the Mock case, with same affinity and a light increase of average distance between extruders (i.e., as observed in SARS-CoV-2, but to a lesser extent), as shown by the best fitting parameters (Supplementary Fig. 6e) and in full agreement with the experimental reports[4]. Therefore, this suggests that the above discussed re-arrangements, caused by alteration of phase-separation properties, are specifically induced by SARS-CoV-2 and are not observed in other viruses. Taking advantage of MD simulations, we produced an example of 3D structure representing the average TAD in Mock (Fig. 2f, left panel and Supplementary Movie 1)

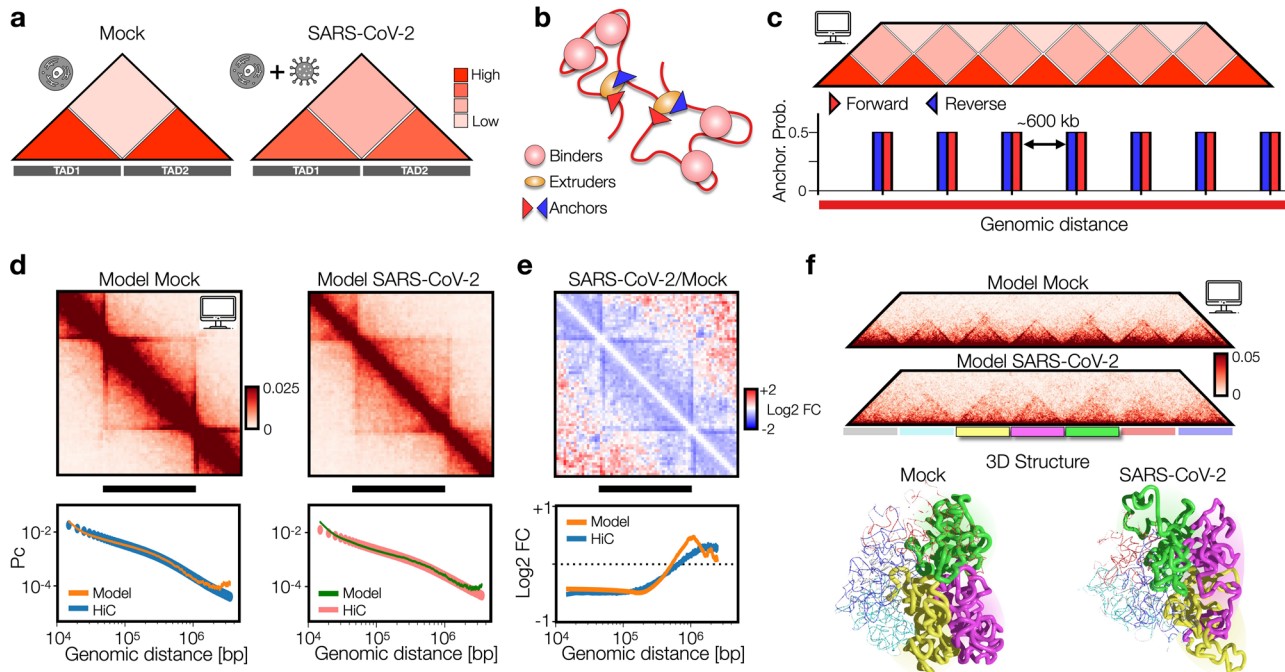

**Fig. 2 | Viral infection impacts loop-extrusion and phase-separation features at TAD level. a** Intra-TAD weakening in SARS-CoV-2 infected genome detected from HiC data[4]. **b** Polymer model of genome organization at TAD scale envisages chromatin loop-extrusion and interactions between chromatin and proteins (binders). **c** TAD boundaries are limited by converging (forward-reverse) anchor points, occurring with a probability (Methods). **d** Average contact maps of TADs from the best model for Mock (left) and SARS-CoV-2 conditions, obtained fitting experimental contact probabilities (Methods). Below, best fit and experimental contact

probabilities. **e** Log2 Fold Change of average contact maps. Below, Log2 Fold Change of contact probabilities in HiC[4] (blue curve) and model (orange curve). Correlation between the curves is $r = 0.82$. **f** Contact maps of best fit models for Mock and SARS-CoV-2 conditions. Below, 3D rendering of polymer structures obtained from MD simulations of Mock and SARS-CoV-2 models. For visualization purposes, only the three central TADs are shown. Source data are provided as a Source Data file.

and SARS-CoV-2 infected (Fig. 2f, right panel and Supplementary Movie 2) conditions, providing an effective and realistic summary of the architectural re-arrangements occurring within and between TADs after the infection. Microscopy experiments could be a possible strategy to observe this structural effect.

Next, to investigate the impact of combined extruders and affinity variation on chromatin compartmentalization, we generalized the above discussed model of TADs by including also A and B compartments (Supplementary Fig. 7a, Methods). When the number of extruders is lowered (we considered ~4-fold reduction), compartment affinities kept fixed, TADs are weakened ($p = 10^{-46}$, one-sided Mann-Whitney U test) and compartmentalization is strengthened (Supplementary Fig. 7b–d Methods), consistent with experimental observation in which depletion of Cohesin increases compartment strength[25,26]. Interestingly, if the same decrease of extruders is coupled with a decrease of the homo-typic affinities, either intra-TAD contacts ($p = 10^{-97}$, one-sided Mann–Whitney U test) and compartmentalization strength are reduced (Supplementary Fig. 7b–d), in agreement with HiC data from SARS-CoV-2 infected cells. Overall, those simulations suggest that SARS-CoV-2 viral infection specifically affects genome organization by altering fundamental physical mechanisms, including loop-extrusion and phase-separation, that shape chromatin structure.

## Structural re-arrangements of interferon response genes (IFN) loci

Next, to understand how the above discussed structural re-arrangement within TADs may affect gene regulation, we modeled real genomic regions relevant in case a viral infection occurs. Specifically, we considered genomic loci containing interferon (IFN) response genes, i.e., genes typically upregulated upon interferon stimulus and that are commonly expressed as response to a viral infection[17].

Importantly, it has been shown that in severe Covid syndromes such genes are not properly expressed[1,27] with consequent alteration in the immunological response of host cell. We considered as first case of study the genomic region spanning 400 kb around the *DDX58* gene (chr9: 32300000-32700000 bp, hg19 assembly, Fig. 3). The *DDX58* locus exhibits the typical re-arrangements caused by SARS-CoV-2 infection, as in Mock case the *DDX58* gene is contained in a well-defined domain limited by convergent CTCF sites (Fig. 3a), whereas in the infected case a general weakening of intra-TAD interactions is observed, although CTCF peaks are mainly unchanged (Fig. 3b). Analogous observations hold for another IFN locus, containing the cluster of *IFIT* genes (chr10: 90900000-91290000 bp) (Supplementary Fig. 8a, b). To quantitatively investigate such re-arrangements, we employed the above-described polymer model combining loop-extrusion and chromatin-protein interactions[16], using experimental CTCF ChIP-seq data[4] to set the probabilities and the positions of the anchor points for extruders[15] and HiC data to optimize the types and the positions of the binding sites[11] (Supplementary Fig. 8a). To this aim, we employed the PRISMR algorithm[11], which infers from the input HiC contact map the number of types of binding sites and their best arrangement along the polymer to fit the input data (Methods). In the *DDX58* locus, the algorithm returned 4 types of binding sites (Fig. 3c, d), while in the *IFIT* locus 5 types have been found (Supplementary Fig. 8c, d). Taking advantage of the results obtained for the polymer model calibrated to simulate the average chromatin behavior at TAD level, we were able to generate, by MD simulations, ensembles of 3D structures accurately capturing the differences in the *DDX58* locus between Mock and SARS-CoV-2 conditions, as shown by the simulated contact maps (Fig. 3c, d) highly correlated with experimental data (Pearson $r > 0.9$, distance corrected $r' = 0.67$, Methods). In addition, the model correctly captures the different contact probability decay

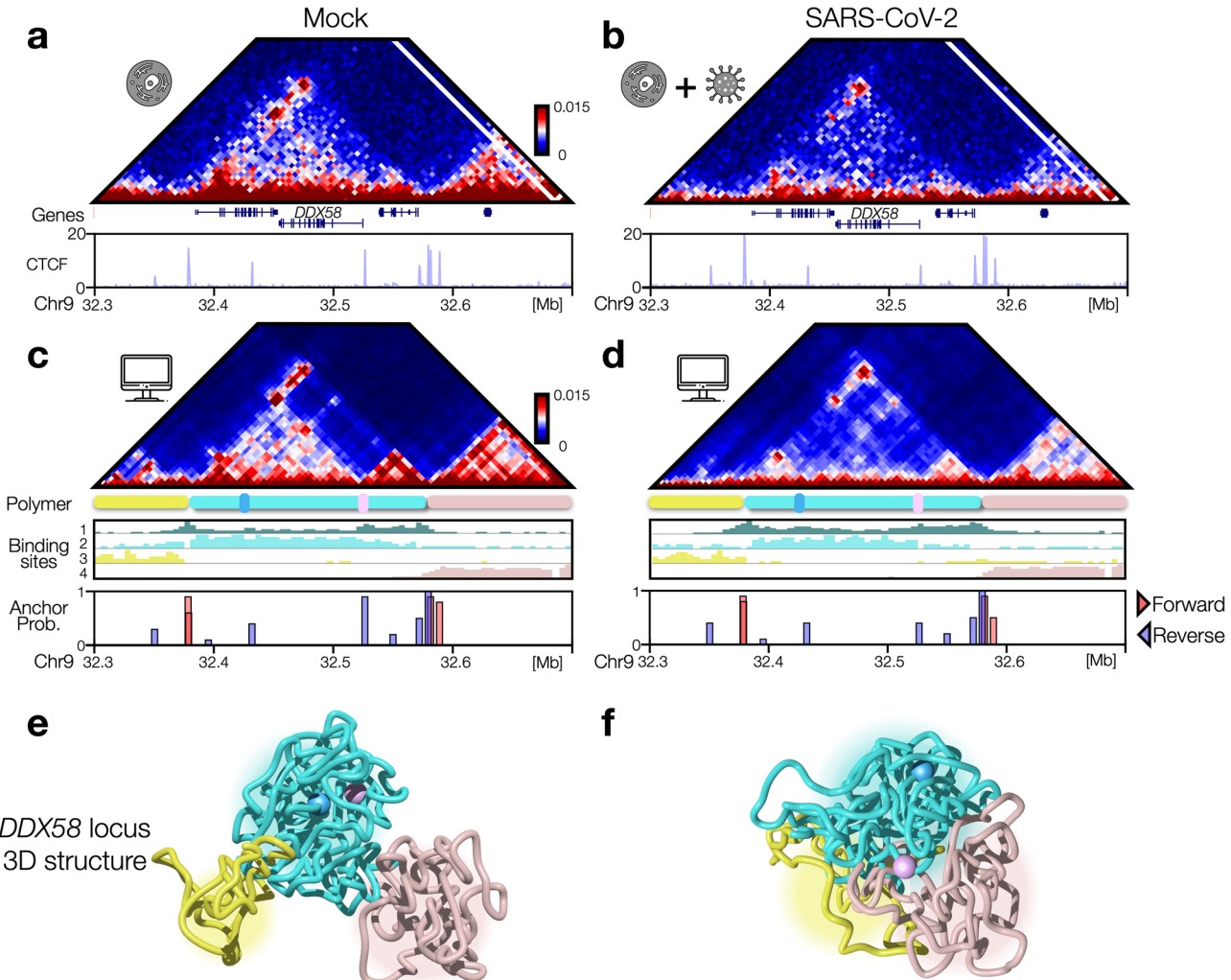

**Fig. 3 | Structural re-arrangements of IFN DDX58 locus. a** Mock HiC data of the genomic region (chr9:32300000-32700000, hg19) centered around the interferon response *DDX58* gene. Below, CTCF signal is shown (data taken from ref. 4). **b** As (**a**), SARS-CoV-2 HiC and CTCF data. **c** Simulated contact map for Mock polymer model. Below, binding sites profile, probability of the anchor points and their orientation (Methods) is shown. **d** As panel c, SARS-CoV-2 model. **e** Example of 3D structure of *DDX58* locus taken from an MD simulation of the Mock model. Different regions are differently colored according to the pattern of the contact maps. Pink and cyan spheres highlight the position of *DDX58* and its enhancer respectively. **f** As (**e**), SARS-CoV-2 model. Source data are provided as a Source Data file.

(Supplementary Fig. 9b), as shown by the Log2 FC curve (Supplementary Fig. 9c, Pearson *r* = 0.81). Analogous results were found for the polymer model of the *IFIT* locus, which returns highly correlated contact maps (Supplementary Fig. 8c, d) and similar contact probability decays (Supplementary Fig. 9d, e). Finally, examples of 3D structures taken from MD simulations (Supplementary Data 1) visually highlight the above-discussed architectural differences, with the *DDX58* and *IFIT* loci organized in distinct, well-defined regions in Mock (Fig. 3e and Supplementary Fig. 8e) while they tend to be less localized and more intermingled in SARS-CoV-2 (Fig. 3f and Supplementary Fig. 8f).

**Single cell 3D structures result highly variable in SARS-CoV-2 infected condition**

The different 3D structures observed in Mock and SARS-CoV-2 prompted us to investigate in more detail the above-discussed architectural differences at the single cell level. To this aim, polymer models offer a powerful tool as they allow to build ensembles of independent 3D structures that mimic single-cell variability[16], experimentally observed e.g., by MERFISH microscopy method[28]. Therefore, leveraging on such feature, we analyzed the population of 3D structures in Mock (Fig. 4a, upper panel) and SARS-CoV-2 (Fig. 4a, bottom panel)

models. First, we focused on the *DDX58* promoter and its validated enhancer[4] (Fig. 4a). By visual inspection of these 3D structures in both conditions, it emerges that *DDX58* promoter and the enhancer tend to be closer in space in Mock with respect to the infected condition, in agreement with HiC data. The distributions of 3D distances between the *DDX58* promoter and its enhancer (Fig. 4b) confirmed this observation, as in Mock it exhibits a lower mean than the infected case (one-sided *t* test $p = 10^{-259}$). Interestingly, the distribution results also more variable in infected cells (st. dev. in SARS-CoV-2 ~30% higher than in Mock), suggesting that the mis-regulation of this gene upon infection is also due to a loss of contact specificity and supporting the scenario by which the viral action changes the binding pattern through alteration of Cohesin and other factors, which in turn causes a general loss of structural coherence in the population of 3D structures. Next, we focused on the architecture of the entire locus and considered the polymer size and shape descriptors[29] (Methods). Again, we find that the estimated volume distribution (Methods) is more variable in SARS-CoV-2 (Fig. 4c upper panel, st. dev. ~30% higher). Conversely, the average anisotropy distribution (Fig. 4c, bottom panel), which measures how asymmetrically the polymer is distributed in space, results lower in SARS-CoV-2 population. Analogous results are found for a-sphericity, another shape descriptor (Methods) measuring the

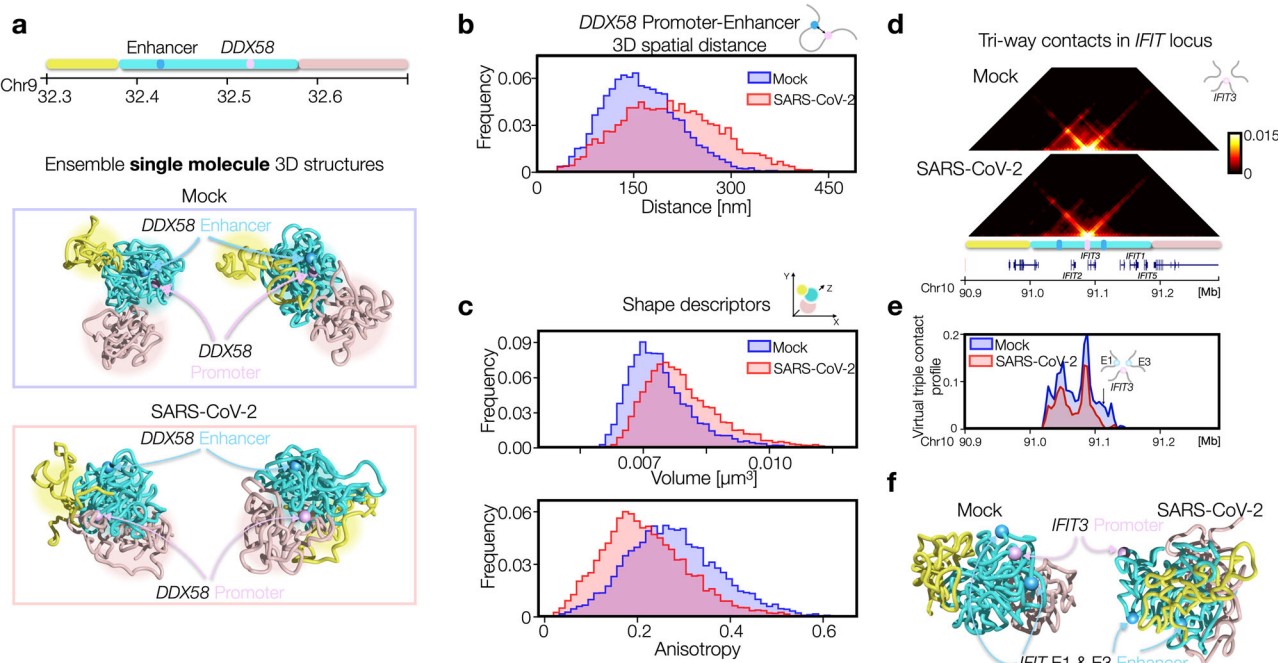

**Fig. 4 | Single cell 3D structures result more variable in SARS-CoV-2 infected condition. a** Examples of 3D structures from the ensemble of single molecule configurations for *DDX58* locus in Mock (top panel) and SARS-CoV-2 (bottom panel) models. *DDX58* promoter and its enhancer are highlighted in pink and cyan respectively. **b** Distributions of 3D distances between *DDX58* promoter and its enhancer. Length scales are estimated by mapping the model in physical units (Methods). Distributions are statistically different (one-sided *t* test, $p = 10^{-259}$). $n = 6000$ for each model. **c** Size and shape descriptors computed from the entire polymer representing the *DDX58* locus. Mock and SARS-CoV-2 models exhibit

different volume (top panel, one-sided *t* test $p = 10^{-240}$) and anisotropy (bottom panel, one-sided *t* test $p = 10^{-240}$) distributions. $n = 6000$ for each model. **d** Simulated matrices of triple contacts using *IFIT3* as point of view in Mock (top panel) and SARS-CoV-2 (bottom panel) models. **e** Virtual triple contact profile fixing enhancer 1 and *IFIT3* as point of views. Arrow indicates probability of E1-*IFIT3*-E3 triple. **f** Examples of 3D structures from the ensemble of single molecule configurations for *IFIT* locus in Mock (left) and SARS-CoV-2 (right) models. Positions of *IFIT3* and enhancers E1 and E3 are highlighted in pink and cyan respectively. Source data are provided as a Source Data file.

deviation from a spherical geometry. Those results are consistent with the results of the previous section, where we observed increased inter-TADs contacts and less localization observed which make the polymer more homogeneous and spherical in SARS-CoV-2 model.

Next, we investigated whether the infected model may exhibit differences on higher-order contacts. To this aim, we focused on the cluster of *IFIT* genes, where we considered the probability of three-way contacts[30,31] using as point of view *IFIT3* gene, located in the center of the *IFIT* TAD (Fig. 4d, Methods). We find that in SARS-CoV-2 model three-way contacts result consistently reduced (Fig. 4d) although weak, long-range events appear. By fixing the enhancer 1 (E1) as other point of view we generated a virtual three-way profile involving E1 and *IFIT3* (Fig. 4e), which clearly highlights specific three-way contacts, as the triplet involving E1-*IFIT3*-E3 (arrow in Fig. 4e) whose frequency in Mock case results statistically higher than in control triplets ($p = 3*10^{-9}$, one-sided Mann–Whitney U test, Methods). In addition, the frequency of this triplet is reduced in SARS-CoV-2 model ($p = 7*10^{-5}$, one-sided Mann-Whitney U test, Methods). This suggests that the mis-regulation may also be due to an alteration of contact network within the regulatory hub, consistent with other recent observations whereby the olfactory hubs are disrupted/perturbed after SARS-CoV-2 infection[5]. Finally, examples of 3D structures of the *IFIT* locus in Mock (Fig. 4f, left panel) and SARS-CoV-2 (Fig. 4f, right panel) conditions provide a visual summary of the discussed results.

### Time dynamics of 3D contacts is highly variable in SARS-CoV-2 infected condition

Next, we investigated the mechanism leading to the different structural variability observed in Mock and SARS-CoV-2 models. To this aim, we considered the population of independent time trajectories (Methods) of the polymer and analyzed the dynamics in both

conditions (Fig. 5a) at equilibrium (Methods). We focused again on the *DDX58*-enhancer distance (Fig. 5b) and the above-discussed polymer shape descriptors, i.e., anisotropy (Supplementary Fig. 10a, left panel) and a-sphericity (Supplementary Fig. 10a, right panel). As expected, the distance trajectories in the Mock model appear fluctuating around average values lower than the SARS-CoV-2 model, as also confirmed by the distributions of the average distance over different time trajectories (Fig. 5c, upper panel, *t* test $p = 10^{-15}$, Methods). Same analysis for anisotropy (Supplementary Fig. 10b, left panel) and a-sphericity (Supplementary Fig. 10b, right panel) reveals instead a specular behavior, in agreement with the observations of the previous section. Interestingly, it emerges also that the time trajectories in SARS-CoV-2 model are more fluctuating, as shown by the distribution of the standard deviations of the distance in time (Fig. 5c, lower panel). For the shape descriptors Mock and SARS-CoV-2 models exhibit similar deviations from the average value during time (Supplementary Fig. 10c). Analogously, multiple co-localization events (named co-occurrences, Methods) in *IFIT* locus, involving *IFIT3* and two enhancers, tend to be less frequent in time in SARS-CoV-2 model dynamics (Supplementary Fig. 10d). These results suggest that SARS-CoV-2 could affect the stability of contacts between regulatory elements. To support this conclusion, we analyzed in more detail the *DDX58*-enhancer distance time dynamics by considering shorter time scales at higher time resolution (Fig. 5d, Methods). We generated time trajectories to follow a smooth evolution of gene-enhancer distance, in Mock (Supplementary Movie 3) and SARS-CoV-2 (Supplementary Movie 4) models. In this way, we were able to estimate a contact time τ, i.e., how long the gene and the enhancer spend in contact (Fig. 5e, Methods). Importantly, we find that the distribution of contact times tends to be significantly lower in SARS-CoV-2 model (Fig. 5f, *t* test $p = 2*10^{-4}$), with an approximately 1.5-fold reduction of the average

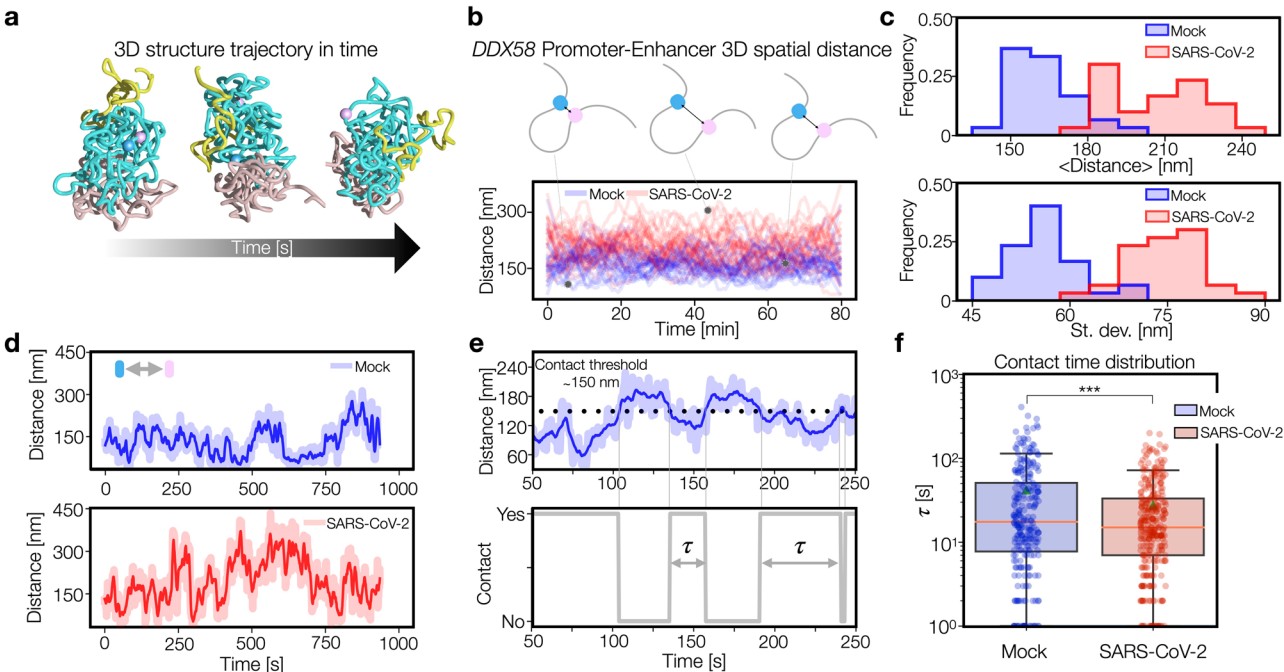

**Fig. 5 | Time dynamics of 3D contacts is more variable in SARS-CoV-2.**
**a** Evolution of 3D structure during time. **b** 3D distance trajectories between *DDX58* promoter and its enhancer in Mock (blue curves) and SARS-CoV-2 (red) models. **c** Distributions of average distance (top) and standard deviation (bottom) computed over independent time trajectories shown in panel b. In both cases, Mock is statistically different from SARS-CoV-2 model ($p = 10^{-15}$ and $10^{-18}$ respectively, one-sided *t* test). $n = 30$ independent trajectories for each model. **d** Examples of 3D distance trajectories at higher time resolution in Mock and SARS-CoV-2 models. Darker lines are smoothed curves (Methods). **e** Given the 3D distance trajectory

(top) we define the contact time $\tau$ (bottom) as the time period spent below the contact threshold, here set to 150 nm (Methods). **f** Boxplots showing the distribution of contact time $\tau$ in Mock (blue) and SARS-CoV-2 (red) models. Distributions have statistically different averages ($p = 2 \times 10^{-4}$, one-sided *t* test). $n = 272$ for Mock and $n = 316$ for SARS-CoV-2 infected model. The centre lines represent medians; triangles represent averages; box limits indicate the 25th and 75th percentiles; and whiskers extend 1.5 times the IQR from the 25th and 75th percentiles. Source data are provided as a Source Data file.

contact time. Analogously, we considered the distribution of time intervals between contacts (Supplementary Fig. 10e) and found that its average exhibits an approximately 1.6-fold increase in SARS-CoV-2 condition (one-sided *t* test $p = 3 \times 10^{-4}$), suggesting that alteration of either contact times and frequencies similarly contribute to the changes in the HiC map observed in infected condition.

Taken together, those results point toward a scenario where the mis-regulation of IFN genes observed in SARS-CoV-2 could be imputed to a decreased contact stability between genes and their regulatory elements. It is worth to stress that, although in-silico generated, the distance dynamics obtained by these polymer models represent a good proxy of real trajectories, as shown in recent studies[32] and are therefore suitable quantities for experimental testing through e.g., live cell imaging[33].

**Chromatin re-arrangements in SARS-CoV-2 infection correlate with a combination of changes of CTCF and histone marks**
Next, to understand the link between the architectural re-arrangements encoded in HiC data and molecular factors, we investigated the relationship between binding sites and epigenetics marks, such as CTCF and histone modifications. In this way, we could assign a biological identity to the binding sites found from HiC data[11,34] and mechanistically interpret the changes in such associations occurring upon viral infection. To this aim, we made a cross-correlation analysis (Methods) between the binding site profiles of the model and different available epigenetic marks at *DDX58* (Fig. 6) and *IFIT* (Supplementary Fig. 11) loci, in Mock and SARS-CoV-2 conditions. In Mock, we find (Fig. 6a, right panel) a clear, strong correlation between CTCF and RAD21 with binding site type #1, likely highlighting an important role for LE mechanism in shaping the central domain containing the *DDX58* gene, but we also observe a significative correlation with RNAPolII

(RPB1) and H3K4me3, in agreement with the view of a combinatorial action of different factors in shaping chromatin organization[34]. In addition, it emerges a clear association between the flanking binding sites (#3 and #4) to H3K27me3 and H3K9me3 respectively (Fig. 6b, left panel). In SARS-CoV-2 model the distribution of binding sites exhibits, in general, a similar profile (Fig. 6a, left panel) but a richer pattern of (less strong) correlations is found (Fig. 6b, right panel). In particular, we could identify the most significant changes in such correlations by using a control set of randomly permuted polymers (Methods) and found that they involve CTCF ($p = 0.047$) and RAD21 ($p = 0.065$, generally reduced), which become associated with multiple types (#1 and #3) as well as H3K27ac ($p = 0.033$), which exhibits a general reduction too[4]. Analogous considerations hold for *IFIT* locus where changes in correlations involve CTCF, RPB1 and H3K4me3 (Supplementary Fig. 11), although they result much less significant ($p > 0.1$). In general, those results support the proposed mechanism[4] by which an alteration of LE activity upon infection coupled with changes in the epigenetic signatures of activity produces an altered expression of IFN genes with a consequent poor response to the infection.

## Discussion
In this work, we investigated how SARS-CoV-2 infection alters the 3D organization of chromatin in the host cell at multiple length scales, ranging from few kilobases to several Mbs and involving different structural entities, as A/B compartments, TADs and gene-enhancer loops. To this aim, we employed models from polymer physics and MD simulations widely used to study chromatin organization[13,18,35]. We showed that a simple block copolymer including just homo-typic and hetero-typic interactions is overall able to describe the A-compartment weakening and A-B mixing detected from HiC data in SARS-CoV-2

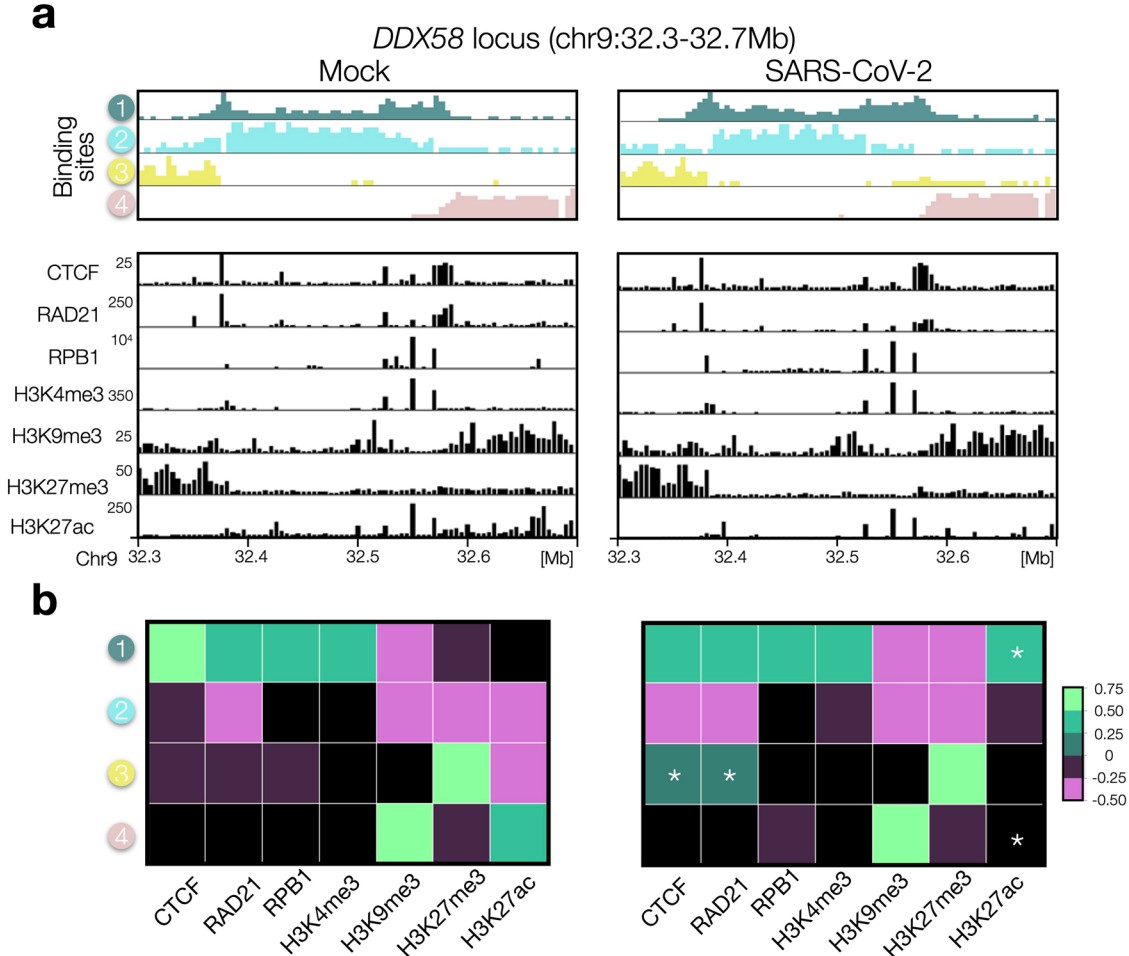

**Fig. 6 | Chromatin re-arrangements correlates with a combination of changes of CTCF and histone marks at DDX58 locus. a** Top panel: distribution of binding sites obtained from HiC data in Mock (left) and SARS-CoV-2 (right) conditions. Bottom panel: different epigenetic marks (data from ref. [4]). In SARS-CoV-2 relevant reductions of RAD21 and H3K27ac are observed. **b** Cross-correlation analysis between binding site profiles and epigenetic marks. Significative correlations (Methods) at *DDX58* locus in Mock (left) and SARS-CoV-2 (right) models are shown. Asterisks indicate strong changes of the correlation between Mock and SARS-CoV-2 infected conditions (Methods). Source data are provided as a Source Data file.

infected cells, by remodulating A-A affinities in an unbalanced A/B compartment model. Of course, more complicated descriptions of compartmentalization are possible and could include other mechanisms known to play a role for chromatin structure, such as interaction with nuclear envelope[19,36]. At TAD level, we find that a combined reduction of loop extrusion activity (modeled as a reduction of extruders) together with an alteration of phase-separation properties (modeled as a reduction of chromatin-protein affinities) potentially explain the weakening of intra-TAD interactions observed in HiC data[4]. Interestingly, a model calibrated from HiC data in host cells infected with virus[4] HCoV-OC43, another human coronavirus causing common cold, has a slightly reduced loop-extrusion activity with respect the Mock case but keeps unchanged protein affinities with chromatin, suggesting that the capacity of altering this phase-separation properties is a peculiar feature of SARS-CoV-2 model and it is not triggered by defense mechanisms of the host cell. In addition, a model including TADs and A/B compartments confirmed this scenario, as simultaneous alteration of loop-extrusion and phase-separation can lead to intra-TAD weakening and a general decrease of compartmentalization strength, as observed in SARS-CoV-2 infected condition. We then investigated the link between chromatin re-arrangement and the regulation of genes involved in the antiviral response (IFN genes) which are mis-regulated upon SARS-CoV-2 infection[1]. Polymer models of genomic loci containing *DDX58* and *IFIT* genes highlighted a higher

degree of variability in the ensemble of single-molecule conformations of SARS-CoV-2 models. This variability is in turn related to a noisier and less stable time of contact dynamics, suggesting that SARS-CoV-2 infection reduces specificity and structural stability of regulatory contacts. Analysis of epigenetic association with the polymer models reveals changes with factors not only limited to Cohesin and CTCF, consistent with the above depicted scenario where SARS-CoV-2 infection alters multiple physical mechanisms shaping chromatin organization of the host cell.

In order to understand the molecular causes leading to the above-discussed re-arrangements, by means of direct or undirect mechanisms, it would be interesting to integrate in the polymer model the existence of specific molecular factors encoded by the virus known to perturb the host cell, as highlighted by recent experiments showing that viral proteins can alter the cell epigenome[2] (ORF8) or interact with other proteins of the infected cell[37]. In this regard, it is worth to mention that other viruses are capable of re-structuring genome organization through the transcription of their proteins, such as NS1 from influenza A virus (IAV)[38]. The above outlined strategy, based on polymer models combined with experimental data, could be relevant to test the effects of specific proteins on the physical mechanisms shaping chromatin architecture (e.g., phase-separation) and therefore be helpful in the identification of molecular targets for therapeutics purposes.

In general, exploring the link between viral infection and chromatin architecture can be extremely insightful to understand virus action on host cell at the level of gene regulation. To this aim, polymer models turn out to be valuable tool as they offer an unbiased, predictive approach to connect different aspects relevant for genome organization and function[39], including single-cell variability, dynamics between regulatory elements and research of therapeutic targets.

## Methods
We use polymer physics models to study chromatin re-organization of host cell genome infected by SARS-CoV-2. We consider recently published HiC data[4] in control condition, i.e., not infected human A549 cells expressing ACE2 (referred to as Mock) and in human A549 cells expressing ACE2 at 24-hour post SARS-CoV-2 infection.

### Polymer model of A/B compartment
To simulate A and B compartment we employed the Strings and Binders Switch[18] (SBS) model, in which a chromatin filament is modeled by a string made of $N$ beads that can interact with different, specific binding factors populating the surrounding environment. We used a polymer made of $N = 1000$ beads, divided in equally sized blocks of 75 beads, schematically colored in green and red (Fig. 1) and representing, without loss of generality, A and B compartment respectively. Assuming, e.g., a genomic content of 100 kb per bead, the polymer represents a region of 100 Mb divided in 12 compartments large 7.5 Mb each, in line with average size of A/B compartments[40]. Homo-typic affinities $E_{A-A}$ and $E_{B-B}$ between binding sites and cognate binders, which mediate intra-compartment interactions (i.e., between A-A and B-B), are taken in the range $3.2-3.4K_BT$. In Fig. 1 and Supplementary Figs. 1 and 3 we show affinities normalized with respect the background hetero-typic interaction (A-B and B-A), which is taken $E_{A-B} = 3.1K_BT$ and kept constant in all the simulations for sake of simplicity. To test the generality of our results, we considered also models with constant homo-typic affinities ($E_{A-A} = E_{B-B}$, for sake of simplicity) and variable hetero-typic affinity $E_{A-B}$ taken in the range $3.0-3.2K_BT$. As before, in Supplementary Figs. 2 and 4 we show affinities normalized with respect $E_{A-B}$. Finally, binder concentration is taken above coil-globule transition threshold[13], so to ensure phase-separation of compartments.

### Polymer model of real chromosomes
To simulate chromosome 11 (Supplementary Fig. 5), we used the 1st eigenvector from the Principal Component Analysis (PCA) applied to HiC data in Mock condition at 100 kb resolution, using the function *eigs_cis* from *cooltools* package[23]. GC content was used to identify A and B compartments. We used a polymer made of $N = 1353$ beads, having 100 kb of genomic content. Beads of type A or type B were assigned based on the compartment profile, as shown in Supplementary Fig. 5a. When the eigenvector was not defined, A/B beads were assigned by simple interpolation around those sites (Supplementary Fig. 5a). As before, homo-typic affinities $E_{A-A}$ and $E_{B-B}$ are taken in the range $3.2-3.4K_BT$ with hetero-typic $E_{A-B} = 3.1K_BT$ kept fixed.

### Polymer model of TADs
To simulate polymer models of TADs we considered again the above mentioned SBS model combined with loop extrusion[14,15] (LE), following a previously described implementation[10]. Specifically, we use a simple homopolymer made of $N = 1000$ beads with one type of binder, as shown in Fig. 2b. Bead-binder interaction affinity is taken in the range $3.1-3.8K_BT$, binder concentration is accordingly taken high enough to ensure coil-globule phase-transition and, as before, it is kept constant for sake of simplicity. Anchor points for the loop extruding factors (LEfs) are all bi-directional (i.e., forward and reverse) and are regularly placed along the polymer every 120 beads, occurring with a probability equal to 0.5, as shown in Fig. 2c. We assume a 5 kb genomic content per

bead, so to obtain an average TAD size ~600 kb, i.e., similar to the average TAD size measured in Mock HiC data[4]. Average distances among extruders (we refer to as LEf separation in Supplementary Fig. 6d, e), proportional to the inverse of their total number, is taken in the range 60−500 kb, consistent with previous reports[15]. Extruders lifetime, which is in turn related to the processivity (i.e., the average length of an extruded loop, see *Molecular Dynamics simulation details*), is taken high enough to allow the formation of TADs and loops (500 kb) in the contact maps and kept constant for sake of simplicity. Results were not significantly affected by changes of this parameter. Analogously, polymer model including TADs and A/B compartments (Supplementary Fig. 7) is made of $N = 1000$ beads with four equally sized compartments in the sequence A-B-A-B (250 beads each), with 5 TADs in each compartment (50 beads each). By assuming 10 kb of genomic content for each bead, we simulate approximately 10 Mb. Again, homo-typic affinities $E_{A-A}$ and $E_{B-B}$ are taken in the range $3.2-3.4K_BT$, hetero-typic affinity $E_{A-B} = 3.1K_BT$ kept fixed. In this case, we considered only balanced affinities ($E_{A-A} = E_{B-B}$) for sake of simplicity. Loop extrusion parameters are similar to the above described model of TADs, with average distance among extruders taken in the range 100−1000 kb (Supplementary Fig. 7a) and similar processivity.

### Polymer model of interferon response genes DDX58 and IFIT
To simulate *DDX58* (chr9:32300000-32700000, hg19) and *IFIT* (chr10:90900000-91290000, hg19) loci, we used the previously described hybrid model (SBS + LE), informed with experimental data to find the binding sites along the polymer, as schematically shown in Supplementary Fig. 9a. Binding sites have been obtained with the PRISMR algorithm[11], using as input HiC data in Mock and SARS-CoV-2 conditions at 5 kb resolution. In general, starting from a contact map of a generic genomic locus, this algorithm returns the minimum number of different types of binding sites (represented by different colors) and their position along the chain in order to best explain the input data. This occurs through an iterative Simulated Annealing Monte Carlo optimization procedure that minimizes a cost function made of two terms: the first one takes into account the difference between HiC and model predicted contact matrices, while the second is proportional to the total number of model binding sites so to penalize the presence of too many of them[11]. In this way, the model optimizes the similarity with the input HiC data, while avoiding overfitting. Here, 4 types of binding sites for *DDX58* (Fig. 3) and 5 types for *IFIT* (Supplementary Fig. 8) locus have been found, in line with similar polymer models of real loci[10], and an inert type not shown in the diagrams for sake of simplicity. As the single bead contains 500 bp, we used polymers of $N = 900$ beads for *DDX58* locus and $N = 880$ beads for *IFIT* locus (we added inert tails of 50 beads on both sides to control boundary effects). Homo-typic bead-binder attractive interactions were taken in the range $2.3-2.9K_BT$, SARS-CoV-2 model simulated with lower affinity with respect the Mock model, consistently with the results obtained in the previously discussed model of TADs. In addition, a general, constant hetero-typic interaction is also used[16]. Anchor points have been defined using CTCF peaks from ChIP-seq data[4] binned at 500 bp resolution (i.e., the size of a single polymer bead). The presence of an anchor point occurs with a probability proportional to the height of the signal. Orientations of anchor points (Fig. 3 and Supplementary Fig. 8) have been assigned using the FIMO tool in the MEME suite (https://meme-suite.org/meme/) fed with CTCF binding motif (JASPAR database)[10,11]. If multiple matches occurred within the 500 bp window, the most likely was taken (i.e., the FIMO hit with lowest $p$ value). Processivity was taken in the range 150−400 kb to ensure the formation of the loops in the maps (Fig. 3 and Supplementary Fig. 8). Separation among extruders is taken in the range 50−100 kb, with SARS-CoV-2 case simulated with halved density with respect to the Mock case, again consistently with the results found from the previously described model of TADs.

## Molecular dynamics simulations details

All previously described polymer models have been explored using classical Molecular Dynamics simulations[41]. In general, chromatin fiber is a standard bead on a string chain and binders are simple spherical particles, both with same diameter $\sigma = 1$ and mass $m = 1$, expressed in dimensionless units. Excluded volume effects of beads and binders are taken into account using a truncated, purely repulsive Lennard-Jones (LJ) potential[41], with energy unit $K_B T$, $T$ temperature and $K_B$ Boltzmann constant. Consecutive beads are linked by FENE bonds[41], with maximum length $R_0 = 1.6\sigma$ and spring constant $K_{FENE} = 30 K_B T/\sigma^2$. Bead-binder attractive interactions are modeled using a short-range, truncated LJ potential: $V_{LJ}(r) = 4\varepsilon[(\frac{\sigma}{r})^{12} - (\frac{\sigma}{r})^6 - (\frac{\sigma}{r_{int}})^{12} + (\frac{\sigma}{r_{int}})^6]$ for $r < r_{int}$ and 0 otherwise, where $r$ is the distance between particle centers and $\varepsilon$, sampled in the range $(8-12)K_B T$, regulates the interaction intensity. Specific parameters are: $r_{int} = 1.3\sigma$ for compartment and TAD models (Figs. 1 and 2), $r_{int} = 2.5\sigma$ for specific interaction in *DDX58* and *IFIT* loci models (Fig. 3 and Supplementary Fig. 8). Unless differently stated, we always show binding affinities corresponding to the minimum of $V_{LJ}$. Length scales are mapped in physical units (Figs. 4 and 5) through the relation[18] $\sigma = (\frac{g}{G})^{1/3} D$ where $D$ is nuclear diameter (7 μm), $G$ is the nuclear genomic content (6.6 Gbp) and $g$ is genomic content of a single bead (500 bp), which returns $\sigma \sim 30$ nm for the models of *DDX58* and *IFIT* loci, in line with previous estimates from analogous polymer models[30].

The system evolves according the Langevin equation[42] with standard parameters[41], i.e., friction coefficient $\zeta = 0.5$ and temperature $T = 1$, in dimensionless units. We used an integration timestep $dt = 0.01$. Integration has been performed with a Velocity Verlet algorithm using the publicly available HOOMD software[43]. Simulations are performed in a cubic box (linear size $L = 50\sigma$ in real loci models) with boundary periodic conditions to take into account finite size effects. For each parameter setting we performed up to 30 independent simulations. Polymer configurations are initialized as standard Self-Avoiding-Walk (SAW) states, binders are randomly located in the simulation box and then equilibrated up to $5*10^7$ timesteps. Configurations have been sampled up to the equilibrium sampling frequency every $5*10^4$ timesteps, except for the simulations shown in Fig. 5d, e, f, where frames were sampled every $10^3$ timesteps. Quantities obtained from the entire population of single-molecule 3D configurations (Fig. 4b, c) are shown as histograms. Analogously, histograms of averages and standard deviations of time trajectories, shown in Fig. 5c and Supplementary Fig. 10b–d, are computed over 30 independent trajectories. Timescales shown in Fig. 5 are estimated by using the relation[13] $\tau = \eta(6\pi\sigma^3/\varepsilon)$, where $\varepsilon = 1 K_B T$ is the energy scale and $\eta$ the viscosity; assuming $T = 300$ K and $\eta = 0.2$ cP, we obtain an estimated timescale $\tau \sim 0.5$ ms, again in line with previous studies[30].

Loop extrusion process is implemented largely following previous descriptions[15] and is integrated in the above-described MD simulations, in a model combining both phase-separation and loop extrusion mechanisms. Basically, loop extruding factors are modeled as harmonic springs with elastic constant $K_{spring} = 10 K_B T/\sigma^2$ and equilibrium distance $r_{eq} = 1.1\sigma$. Extruding factors slide along the polymer every 500 MD timesteps by moving the spring from the bead pair $(i, j)$ to $(i-1, j+1)$. Extruders can stochastically detach from the polymer with a rate $k_{off}$, which is related to the processivity through the relation[44] $proc = 2g/k_{off}$, $g$ the above defined genomic content per bead. When an extruder detaches, a new one is replaced along the polymer in a random position, so to keep a constant number of extruders. An extruder halts its motion when it meets oppositely directed anchor points or when it meets another extruder during the sliding[15], since they cannot pass through each other.

The codes of the above-described models, i.e., performing MD simulations of A/B compartments, TADs and real genomic loci, where SBS and LE are combined, have been adapted from the software[10] at

GitHub link (https://github.com/ehsanirani/PhaseSeparation-LoopExtrusion-MD) and are available as Supplementary Software 1.

## Analysis of contact maps, saddle-plots and contact probability

Contact maps were computed from the ensemble of 3D polymer structures by setting a distance threshold $A$ and defining a contact if $d_{i,j} < A\sigma$, where $d_{i,j}$ is the Euclidean distance between beads $i$ and $j$ and $A$ taken is taken the range 2–3.5. For each polymer conformation, we calculate the associated contact map and then aggregate all the maps of the ensemble of structures for a fixed choice of parameters. All the simulated maps correspond to the entire polymer, except Fig. 1c, Supplementary Figs. 1a, 2 and 3a, where an average among three consecutive sub-matrices is shown for presentation purposes. Analogously, triplet matrices (Fig. 4d) are computed from a simple generalization of the pairwise calculation[30]. We first fix a specific point of view (i.e., the gene *IFIT3*, Fig. 4d) and identify it on the polymer (e.g., bead $i$). Then, from each polymer conformation we call a triple contact of bead $i$ with other beads (e.g., $j$ and $k$) if their mutual Euclidean distance is lower than the threshold or, more formally, if $d_{i,j}$ & $d_{j,k}$ & $d_{k,i}$ < $5\sigma$. Then, we iterate over all possible $j$ and $k$ indexes to obtain a triplet matrix of single polymer conformation. Those matrices are then aggregated to generate a triplet frequency matrix. Statistical significance of the triplet frequency involving E1-*IFIT3*-E3 (Fig. 4e) is estimated by comparing the distribution of triple contacts from the population of single conformations with the distribution of control triplets located 100 kb downstream the *IFIT3* promoter and preserving the relative genomic distance. Saddle-plots (Fig. 1e, Supplementary Figs. 1b, 3b, 4a, 5e) have been computed using the *cooltools* package[23] of the *cooler* tool to analyze HiC data[22]. Briefly, we first converted the simulated maps in *cool* format using the *create_cooler* function, then we called A/B compartments and then used the *saddle* function with default number of bins (i.e., 50). Analogously, we performed the same analysis on HiC data[4] (80 kb resolution) in Mock and SARS-CoV-2 infected conditions to generate the Log2-FC matrix in Fig. 1e. Best polymer model for A/B compartments in Fig. 1d was found by considering linear combinations of simulated saddle-plots and minimizing the sum of the entry-by-entry square difference between the model and experimental saddle-plots. As we considered combinations of four matrices, 2 with unbalanced (i.e., $E_{A-A} < E_{B-B}$) and 2 with balanced (i.e., $E_{A-A} = E_{B-B}$) homo-typic affinities, the procedure finds the best four coefficients (Fig. 2d, bottom panels), their sum constrained to 1. We verified that by minimizing other quantities (e.g., the $\chi^2$) analogous results were found. Similarly, best model in Supplementary Fig. 4a was found by considering combinations of two matrices with different hetero-typic affinities $E_{A-B}$ and balanced homo-typic affinities $E_{A-A} = E_{B-B}$ (specifically, we combined models with $E_{A-A}/E_{A-B} = 1.06$ and 1.08). On the same line, best model for chromosome 11 (Supplementary Fig. 5b, c, d, e) is a combination of two matrices with different homo-typic affinities (one balanced $E_{A-A} = E_{B-B}$ and one unbalanced $E_{A-A} < E_{B-B}$, hetero-typic $E_{A-B}$ kept fixed). In this case, we could fit simulated and experimental 1st eigenvector profile (Supplementary Fig. 5b, c), which returns the same best coefficient (Supplementary Fig. 5d, e) obtained by performing the above-described fit using saddle-plots. Compartment strength (Supplementary Fig. 7c) has been computed using the *saddle_strength* function from the *cooltools* package[23]. Contact probabilities shown in Fig. 2d and Supplementary Fig. 6c were computed from the previously defined contact maps by taking the average value of each diagonal. Curves are then multiplied by a coefficient (equal in Mock and SARS-CoV-2 models) to map the simulated values into the experimental range. Best model for TADs was found by considering the best linear combination of contact probabilities for different model parameters (i.e., average LEf separation and interaction affinity) and then minimizing the $\chi^2$ with experimental curve in the range 15 kbp–2.5 Mb (Mock or SARS-CoV-2 conditions), so to take into account the wide range of variability of the contact

probability. To test the best description in terms of affinities, we consider combinations of four curves, two with affinities 3.1 and $3.8K_BT$ (LEf separation fixed) and two with same affinities but with no LEfs (Supplementary Fig. 6d and 6e). Therefore, the fit returns four coefficients, their sum constrained to 1, representing the amount of each curve in the best model.

## Gene-enhancer distance, shape descriptors and structural variability

3D distances between *DDX58* gene and its enhancer E has been simply obtained by calculating the Euclidean 3D distance from a 3D structure. Smoothing of 3D distance trajectories shown in Fig. 5d and e has been done with a standard 1st order polynomial computed by use of *signal.savgol_filter* function from the Python package *scipy*. Shape descriptors were computed using standard formula used in polymer physics field. We first computed the gyration tensor $\mathbf{G}$[29], whose entries are $G_{\alpha,\beta} = \frac{1}{N}(\sum_i^N (x_{\alpha,i} - x_{\alpha,CM})(x_{\beta,i} - x_{\beta,CM}))$, where $\alpha, \beta \in \{0,1,2\}$ are component indexes, $x_i$ is the vector position of bead $i$, $x_{CM}$ is the vector position of the polymer center of mass and $N$ number of polymer beads. Then, by diagonalizing this tensor, we obtained the three eigenvalues $\lambda_1, \lambda_2$ and $\lambda_3$, sorted in ascending order. Anisotropy (Fig. 4c and Supplementary Fig. 10a–c) is defined as[29] $1 - 3(\lambda_1\lambda_2 + \lambda_2\lambda_3 + \lambda_3\lambda_1)/(\lambda_1 + \lambda_2 + \lambda_3)^2$ and reflects the symmetry of a polymer conformation. Analogously, asphericity (Supplementary Fig. 10a–c) is defined as[29] $(\lambda_1 - (\lambda_2 + \lambda_3)/2)$ and measures the deviation from a spherical symmetry. Volume of a polymer conformation is estimated by first numerically computing a convex hull from the 3D coordinates of the polymer by use of the *spatial.ConvexHull* function from the Python package *scipy* and then converting this value in physical volume units through the previously estimated length scale.

## Epigenetics signature of binding sites

To investigate the biological nature of the model binding sites, we compared their genomic locations with available epigenetic marks. To this aim, cross correlation analysis (Fig. 6 and Supplementary Fig. 11) has been performed as previously described[11,30]. Epigenetics data (Fig. 6a and Supplementary Fig. 11a, left and right bottom panels) are taken from ref. 4 and have been first binned at 5 kb resolution in order to match the HiC resolution used to infer the binding site profiles. Then we computed the Pearson correlation between each specific binding site and epigenetic profiles, i.e., between the number of binding sites of a specific type (represented by a color in the left and right upper panels of Fig. 6a and Supplementary Fig. 11a) and the epigenetic signal in the corresponding 5 kb bins. Significance of these correlations has been estimated with a random control model generated by bootstrapping 10,000 times the original binding sites position along the locus and re-calculating the correlations. We symmetrically set the bottom 15th percentile and top 85th percentile as significance thresholds, although different thresholds led to similar results. The results are collected in a matrix (Fig. 6b and Supplementary Fig. 11b) where each element is the significant correlation between a specific type and epigenetic mark pair, or zero if the correlation is not significant. Typically, each type correlates with a combination of epigenetic marks, rather than with a specific one. Analogously, *p*-values of the changes in correlation with epigenetic tracks between Mock and SARS-CoV-2 has been estimated by comparing the differences with a control distribution of changes obtained by randomly bootstrapping the original binding sites (one-sided computation on a population of 1000 permutations). The top 4 most significant changes (i.e., the lowest *p*-values) were highlighted in Fig. 6.

## Statistics and reproducibility

No statistical method was used to predetermine sample size. No data were excluded from the analyses. The experiments were not randomized. For all boxplots, the centre lines represent medians; box limits indicate the 25th and 75th percentiles; and whiskers extend 1.5 times the interquartile range (IQR) from the 25th and 75th percentiles. Mann–Whitney U test and t-test were commonly used to compare distributions; $p < 0.05$ was considered significant ($*p < 0.05$; $**p < 0.01$; $***p < 0.001$).

## Reporting summary

Further information on research design is available in the Nature Portfolio Reporting Summary linked to this article.

## Data availability

Published HiC data[4] and ChIP-seq data[4] used in this work are available at the Gene Expression Omnibus (GEO) database with accession number GSE179184 CTCF binding motif is available from the JASPAR database (matrix profile MA0139.1). Polymer configurations of *DDX58* locus are provided as Supplementary Data 1. The polymer structures generated for the *IFIT* locus and Chr11 will be available from the authors upon request. Please contact Andrea M. Chiariello at andreamaria.chiariello@infn.it. Requests of these data will be answered within approximately two weeks. Source data are provided with this paper.

## Code availability

Codes used to perform simulations presented in this paper are available in the Supplementary Software 1 folder. The software used for Molecular Dynamics simulations is HOOMD, version 2.9.6. Full information and additional documentation are available at the github link: https://github.com/ehsanirani/PhaseSeparation-LoopExtrusion-MD[10].

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

## Acknowledgements

A.M.C. acknowledges "Programma per il Finanziamento della Ricerca di Ateneo Linea B" (FRA) 2020, University of Naples Federico II, CINECA ISCRA Grant ID PhaSSep - HP10C8JWU7. M.N. acknowledges support from the National Institutes of Health Common Fund 4D Nucleome Program grant 5 1UM1HG011585-03, EU H2020 Marie Curie ITN n.813282, PNRR MUR M4C2 CN00000041 "National Center for Gene Therapy and Drugs based on RNA Technology" NextGenerationEU CUP E63C22000940007, MUR PRIN 2022 2022R8YXMR, and computer resources from INFN, CINECA, ENEA CRESCO/ ENEAGRID88 and Ibisco at the University of Naples.

## Author contributions

A.M.C and M.N. designed the project. A.M.C. and A.A. developed the modeling part; A.M.C., A.A., S.B., A.E., and A.F. ran the computer simulations; S.B and A.E. are other equal contributing authors; A.M.C., A.A., S.B., A.E., F.V., A.F. performed data analyses. A.M.C. wrote the manuscript with input from the other authors.

## Competing interests

The authors declare no competing interests.
