## [Peer Review File · Nature Communications]

REVIEWER COMMENTS

Reviewer #1 (Remarks to the Author):

The manuscript by Nicodemi and colleagues provides an elegant theoretical analysis of SARS-CoV-2 induced changes in genome architecture previously described in a human cell line (A549) expressing ectopic Ace2. The experimental report by Wang et al, showed that SARS-CoV-2 infection remodels the host genome resulting in increased mixing of A-B compartments and reduced intra-TAD contacts. The authors of that paper predicted that reduced self-affinity for A compartments, and reduced loop extrusion may explain compartment intermixing and intra-TAD contact reduction respectively. Here, the authors provide a modeling-based approach that confirms these predictions. Thus, from a theorist's perspective, this manuscript is important. The only concerns I have in regards of the tested models, is whether the authors have considered the possibility that viral infection increases A-B affinity, instead of altering the balance of AA/BB affinity, which is their working hypothesis. The second concern, is that previous work with conditional cohesin deletion in vivo has shown increase of compartment strength- how do they reconcile their predictions of reduced loop extrusion as an explanation for intra-TAD contact reduction, with the expectation that compartment strength should increase (which is not what happens in infected cells)?

Beyond these two specific points, I am not convinced that the manuscript pushes the envelope in regards of providing a better understanding of the effect of SARS-CoV-2 infection in genome architecture. Why is the balance between AA and BB affinity altered upon infection? Can the authors make testable predictions into why SARS-CoV-2 and not common Coronavirus have this effect? I understand that it is not simple to obtain answers on these questions, but if the authors could provide reasonable suggestions and hypotheses, the manuscript would improve significantly.

Reviewer #2 (Remarks to the Author):

The paper explores the findings reported in recent publications concerning the alterations in chromatin Topologically Associated Domains (TADs) following infection by SARS-CoV-2 (references 4 and 5). The study employs a model that incorporates two types of nucleosomes, denoted as A and B, with varying affinities for each other. Additionally, it considers a reduction in loop extrusion, which may be associated with an observed decrease in cohesin levels during infection.

One notable aspect is the discussion regarding how SARS-CoV-2 may influence the expression of interferon (IFN) genes through the direct modulation of long-range nucleosome contacts, as mentioned in lines 326-330.

However, a significant concern arises from the ambiguity of the A-B model. It is unclear whether A corresponds to heterochromatin and B to euchromatin. Consequently, it becomes challenging to map the model's predictions to the actual silencing or activation of genes. It remains uncertain whether A-type and B-type nucleosomes are purely abstract concepts and, if so, what purpose they serve. Moreover, the positioning of A and B along a chromosome seems to offer numerous fitting options, raising questions about the meaning of predictions in the current model.

The dynamics of the distance between promoters and enhancers in Figures 4 and 5 do not appear to be validated against experimental data. While the model's predictions may be plausible, they rely on assumptions about the positioning of A and B nucleosomes between and around promoter-enhancer pairs. This raises doubts about the meaning of predictions from these figures.

The inclusion of the CTCF component in the model (Figure 6) presents complexity in understanding its relevance to Hi-C data, especially when the model only encompasses A and B nucleosomes. It remains unclear what insights Figure 6 provides or whether it is necessary for the overall understanding of the research.

In summary, while the model can be adjusted to align with the observed weakening of TADs, its predictive power and utility remain unclear. It is uncertain whether a model that simply weakens all bindings could equally replicate the data for infected cells. Overall, the results may not be compelling enough to warrant publication in a high-profile journal.

Minor points:

1) Clarify the definition of "MOC" in line 80. It is unclear whether "mock" refers to cells infected by SARS-CoV-2 or cells before infection.

2) Regarding DDX58 and IFIT genes, their expression during COVID-19 infection presents ambiguity. It is essential to determine whether the changes in the binding patterns of the DDX58 gene are caused by SARS genes or by the reorganization of cellular defense mechanisms. This distinction will help clarify whether the observed changes represent defects or gains, as suggested in the text around line 257.

3) In the discussion, it might be more critical to specify whether the observed changes are due to viral genes attempting to directly reduce interferon (IFN) production or whether they result from cellular responses aiming to activate IFN genes.

Reviewer #3 (Remarks to the Author):

This paper sought to address a significant issue: the impact of SARS-CoV-2 infection on the host cell's chromatin architecture. The authors effectively highlight gaps in our understanding of the biophysical mechanisms by which the virus infection alters 3D chromatin structure. The study aims to fill this gap by employing polymer physics models and Molecular Dynamics (MD) simulations to study chromatin reorganizations at multiple genomic scales. The paper is ambitious in its scope, aiming to elucidate the mechanistic underpinnings of how SARS-CoV-2 affects chromatin organization and, by extension, gene regulation. It offers insights into the architecture of specific genomic loci linked to the antiviral interferon response, which is of high immunological interest. Some findings by modeling are very interesting and cannot be readily obtained with current experimental methods or datasets, for example, the looping dynamics between genes and enhancers and the three-way interactions. Overall, this study is well-executed, clearly-written, and sets the stage for a potentially impactful study that could provide valuable insights into the epigenomic effects of SARS-CoV-2 infection. However, I have some comments that the authors shall consider addressing before publication:

1). In the first A/B compartment modeling section, the authors employ a simplified block co-polymer model. While this is a starting point, it may not adequately reflect the biophysical properties of a real genome. I suggest using a polymer model that mimics an actual chromosome (or part of it) with real A/B patterns to better support the conclusions.

2). The authors state that balanced A-A/B-B interactions dominate in the mock genome. This seems to contradict the prevailing notion that homotypic interactions and mediating machinery between A-A and B-B compartments are distinct. This point needs justification.

3). Similar to my first comment, the loop extrusion modeling in Fig. 2 appears to use an artificial locus. Given that loop extrusion and phase separation may vary between euchromatin and heterochromatin, it would be beneficial to model these processes using actual chromatin regions.

4). The part of three-way interactions is intriguing. However, it would strengthen the paper if the authors could demonstrate that the frequencies of these interactions are statistically higher than random occurrences based on their modeling data.

5). In the section on enhancer-promoter time dynamics, the authors note that contact time is generally shorter in infected cells. Could the authors also discuss the changes of contact frequencies (intervals between each contact)? And which (contact times or frequencies) contributed more to the changes on bulk contact maps?

6). The binding site analyses in Figure 6 shows graphical minor changes across four types of binding sites. Statistical testing could help clarify the significance of these differences. Additionally, the criteria for determining the number of types of binding sites should be clarified. Would further classification of binding sites show clearer patterns?

7). For the benefit of a broader audience, I recommend that the authors provide more detailed explanations of their analyses in the methods section, particularly concerning the "Epigenetic signature of binding sites" and three-way interactions.

8). Providing a step-by-step notebook for readers to better understand and reproduce the modeling and analyses would be a very valuable addition.

Point-to-point Reply

Reviewer #1

Reviewer: *The manuscript by Nicodemi and colleagues provides an elegant theoretical analysis of SARS-CoV-2 induced changes in genome architecture previously described in a human cell line (A549) expressing ectopic Ace2. The experimental report by Wang et al, showed that SARS-CoV-2 infection remodels the host genome resulting in increased mixing of A-B compartments and reduced intra-TAD contacts. The authors of that paper predicted that reduced self-affinity for A compartments, and reduced loop extrusion may explain compartment intermixing and intra-TAD contact reduction respectively. Here, the authors provide a modeling-based approach that confirms these predictions. Thus, from a theorist's perspective, this manuscript is important.*

We thank the reviewer for finding our manuscript “important”. We feel that we have addressed all her/his concerns, as detailed in the following point-to-point reply.

Reviewer: *The only concerns I have in regards of the tested models, is whether the authors have considered the possibility that viral infection increases A-B affinity, instead of altering the balance of AA/BB affinity, which is their working hypothesis.*

This is an important point. To investigate genome re-structuring at the A/B compartment level, we started, for sake of simplicity, to use a toy model in which the hetero-typic A-B affinities are kept constant and varied homo-typic (A-A and B-B) only. Although simple, with this model we were able to explain A-compartment weakening and enhancement of A-B compartment mixing through re-modulation of those affinities, as shown in Figure 1D. Anyway, following Reviewer’s suggestion, we performed additional Molecular Dynamics simulations considering this time different values of hetero-typic A-B affinities. Specifically, we sampled hetero-typic affinities E_{A-B} in the range 3.0-3.2 $K_B T$ (Figure R1, panel A), and compared them with experimental data. We express affinities as ratio E_{A-A}/E_{A-B} . Interestingly, those simulations revealed that enhanced mixing between A-B compartments observed in SARS-CoV-2 infected cells can be explained by an increase of hetero-typic interaction E_{A-B} , keeping fixed E_{A-A} and E_{B-B} (Figure R1, panel B), which by the way cannot produce the unbalanced weakening of A compartment. Indeed, correlation of the LogFC of the saddle-plots is 0.6 (Figure R1, panel C), lower than the model with unbalanced homo-typic affinities (0.77). Compartment mixing from increase of hetero-typic affinity is clearly shown in examples of 3D structures (Figure R1, panel D).

We discussed this in the revised section “Modelling of chromatin re-structuring in A/B compartments”, in the Methods section “Polymer model of A/B compartment” and added those results in new Extended Figure 1.

Figure R1: Modelling of A/B compartment with different heterotypic affinity. **A:** Contact maps varying hetero-typic affinity E_{A-B} . **B:** Saddle-plots of best fitting models from Mock and SARS-CoV-2 infected case. **C:** Log2 Fold Change of saddles of model vs HiC data (Pearson correlation 0.6). **D:** 3D structures obtained from MD simulations performed with lower (left) and higher (right) E_{A-B} affinity.

Reviewer: The second concern, is that previous work with conditional cohesin deletion in vivo has shown increase of compartment strength- how do they reconcile their predictions of reduced loop extrusion as an explanation for intra-TAD contact reduction, with the expectation that compartment strength should increase (which is not what happens in infected cells)?

We thank the reviewer for this very interesting question, which raises an important point relevant not only in the context of this work but in general for the chromatin architecture field. To investigate this aspect, we made additional MD simulations of a polymer model including A and B compartments (whose formation is driven by phase-separation PS) and loop extrusion (LE), as schematically depicted in Figure R2, panel A. Specifically, we studied the compartmentalization strength in different conditions. As reference level for the model parameters, we employed affinities previously used (i.e. sampled in the range $3.1-3.3K_B T$, Figure 1) and a number of extruding factors compatible with previous studies (Fudenberg, G. et al., *Cell Rep.* **15**, 2038–2049 (2016)) i.e. LEFs average separation approximately 200kb, and in line with the results discussed in Figure 2, which represent our control condition (Figure R1, panel B, left matrix). When the number of extruders is dramatically lowered (about 4-fold reduction), the affinities kept fixed, the saddle-plot analysis reveals strengthened compartmentalization (Figure R2, panel B, middle matrix) along with intra-TAD contact weakening ($p\text{-val}=10^{-46}$, one-sided Mann-Whitney U test), in full agreement with the experimental observation mentioned by the Reviewer by which deletion of Cohesin enhance compartment strength (Schwarzer, W. et al., *Nature* **551**, 51–56 (2017), Rao, S. S. P. et al., *Cell* **171**, 305-320.e24 (2017)). Interestingly, if the same decrease of extruders is coupled with a decrease of the homo-typic affinity A-A and B-B (we acted on both for sake of simplicity), both intraTAD contacts ($p\text{val}=10^{-97}$, one-sided Mann-Whitney U test) and compartmentalization level are reduced (Figure R2, panel B, right matrix), fully consistent with HiC data from SARS-CoV-2 infected cells. Those results are quantitatively highlighted by the compartmentalization strength (*bioRxiv* 2022.10.31.514564 (2022)) profile in the three conditions (Figure R2, panel C). This result suggests therefore a combined action of LE and PS mechanisms and reconciles the apparent contradiction with literature, strengthening even more our results pointing toward a scenario in which the viral action impacts the LE mechanism along with the phase-separation properties of chromatin, perhaps by altering the protein network in the host cell (Gordon, D. E. et al., *Nature* **583**, 459–468 (2020)). We discussed that in the revised version of the paper, section “Modelling of chromatin re-structuring in A/B compartments”, added a new figure in Extended data 1, panels E, F, G and added all the technical details in the revised Methods section “Polymer model of A/B compartment”.

Figure R2. Combined alteration of LE and PS explains compartment weakening in SARS-CoV-2 infected cells. **A:** Polymer model including either LE, driving TAD formation, and PS driving A/B compartment formation. **B:** Saddle-plots corresponding to different system parameters (distance between extruders or, equivalently their number, and affinity). Reduction of extruders

leads to compartment strengthening. Combined reduction of extruders and affinity leads to compartment weakening. **C**: Compartment strength profile for the different model parameters.

Reviewer: Beyond these two specific points, I am not convinced that the manuscript pushes the envelope in regards of providing a better understanding of the effect of SARS-CoV-2 infection in genome architecture. Why is the balance between AA and BB affinity altered upon infection? Can the authors make testable predictions into why SARS-CoV-2 and not common Coronavirus have this effect? I understand that it is not simple to obtain answers on these questions, but if the authors could provide reasonable suggestions and hypotheses, the manuscript would improve significantly.

This is an interesting point too. As stressed by the reviewer, this is a difficult question whose answer goes beyond the aim of this manuscript, which is focused on the effects produced on architecture by SARS-CoV-2. Nevertheless, we tried to provide some clues in this direction. Indeed, in our paper we already show that the re-arrangements are peculiar of SARS-CoV-2, as already shown in Suppl. Fig. S2, panel E, where we applied the same analysis on HiC data from cells infected with the common cold human coronavirus HCoV-OC43 (taken from Wang, R. et al., *Nat. Microbiol.* **8**, 679–694 (2023)) and we found that re-arrangements are much more limited and involve only number of extruders, while affinities are unchanged. This suggests that e.g. testable predictions can involve analysis of phase-separation properties of the protein machinery associated to A-compartment and known to be linked to viral infection. We stressed more this concept in the revised version of the paper.

Reviewer #2

Reviewer: The paper explores the findings reported in recent publications concerning the alterations in chromatin Topologically Associated Domains (TADs) following infection by SARS-CoV-2 (references 4 and 5). The study employs a model that incorporates two types of nucleosomes, denoted as A and B, with varying affinities for each other. Additionally, it considers a reduction in loop extrusion, which may be associated with an observed decrease in cohesin levels during infection.

One notable aspect is the discussion regarding how SARS-CoV-2 may influence the expression of interferon (IFN) genes through the direct modulation of long-range nucleosome contacts, as mentioned in lines 326-330.

We thank the reviewer for finding “notable” aspects in our work.

However, a significant concern arises from the ambiguity of the A-B model. It is unclear whether A corresponds to heterochromatin and B to euchromatin. Consequently, it becomes challenging to map the model's predictions to the actual silencing or activation of genes. It remains uncertain whether A-type and B-type nucleosomes are purely abstract concepts and, if so, what purpose they serve. Moreover, the positioning of A and B along a chromosome seems to offer numerous fitting options, raising questions about the meaning of predictions in the current model.

In this work, we considered chromatin organization at multiple length scales, including A/B compartment (Figure 1), TADs (Figure 2) and specific loci containing IFN genes (Figures 3-6), modelled using HiC data and CTCF ChIP-Seq data (taken from Wang, R. et al., *Nat. Microbiol.* **8**, 679–694 (2023)).

The classification of polymer beads as A/B compartments only relates to a limited part of the paper where the classification of polymer beads in A-type and B-type do not refer to specific chromatin region but are meant to simulate an average chromatin A/B compartmentalization.

Nevertheless, following Reviewer's comment and to test the validity of the results about A/B compartments (Figure 1) on a real case of study, we performed additional MD simulation of a polymer model simulating chromosome 11 (Figure R3), using A and B bead types. Specifically, we used the 1st eigenvector from the PCA analysis of HiC data (100kb resolution) to define A/B

compartments as only ingredient to build the model, then we defined polymer beads of A-type (in green) and B-type (in red) accordingly (Figure R3, panel A). We then applied the same strategy of Figure 1 and generated an ensemble of 3D structures using parameters previously discussed, from which we obtained model contact maps as well as eigenvector profiles from the standard PCA analysis. We find that the model is in good agreement with experimental data in Mock (Figure R3, panel B) and SARS-CoV-2 (Figure R3, panel C) conditions (Pearson correlation $r > 0.9$ and distance corrected correlation $r' \sim 0.5$ in both cases). Also, eigenvectors from PCA analysis on model matrices are highly correlated with experiments (Pearson $r = 0.9$ in both cases). Therefore, those results show that our polymer models are able to recapitulate experimental data even at the A/B compartment level, highlighting the validity and the purpose of our approach. We clarified this important aspect in the revised manuscript, section “*Modelling of chromatin restructuring in A/B compartments*”, in Methods section “*Polymer model of A/B compartments*” and included those results in the new Extended Fig. 2.

Figure R3: Polymer model including A/B compartment profile recapitulates genome organization in Mock and SARS-CoV-2 conditions. **A:** We defined a polymer model based on 1st eigenvector, from which A/B compartments are defined. **B:** Contact maps simulated in Mock condition accurately recapitulates experimental HiC data (Pearson $r = 0.94$, distance corrected $r' = 0.5$). In the middle, 1st eigenvector of the PCA analysis from HiC (up) and model (bottom) contact maps (Pearson $r = 0.89$). **C:** As panel B, for SARS-CoV-2 infected condition.

Reviewer: *The dynamics of the distance between promoters and enhancers in Figures 4 and 5 do not appear to be validated against experimental data. While the model's predictions may be plausible, they rely on assumptions about the positioning of A and B nucleosomes between and around promoter-enhancer pairs. This raises doubts about the meaning of predictions from these figures.*

We clarify that results discussed in Figure 4 and 5 are obtained from a polymer model describing the locus containing the interferon response (IFN) DDX58 gene (chr9: 32300000-32700000bp, hg19) and does not rely on any assumptions about A/B compartments, as it is 400kb in size and it is entirely contained in one compartment, therefore A/B classification of beads is not possible. Analogous considerations hold for locus containing IFIT gene cluster (chr10: 90900000-91290000bp, hg19). In both models, binding sites around the promoter-genes pairs are found following an established computational procedure (PRISMR) that takes as input HiC data and returns the required number of binding types and their position along the polymer (details are extensively described in the reference paper Bianco, S. et al. *Nat. Genet.* **50**, 662–667 (2018)). In addition, we use CTCF ChIP-Seq data to set anchor points for the loop extrusion (Conte, M. et al. *Nat. Commun.* **13**, 1–13 (2022)). We better stressed and clarified these fundamental aspects in the revised version, section “*Structural re-arrangements of interferon response genes (IFN) loci*” and in the Methods section “*Polymer model of interferon response genes DDX58 and IFIT*”.

Regarding the dynamics, we thank the reviewer to consider “*plausible*” our model predictions. It is correct that we do not have validation against experiments of distance distributions in SARS-CoV-2 and we think that experimental data to validate model predictions can be very interesting direction

for further developments of this work. Nevertheless, from previous studies we have evidence that this kind of polymer models provide accurate predictions of distance distributions between gene and regulators (see e.g. recent Chiariello, A. M. et al. *Cell Rep.* **30**, 2125-2135.e5 (2020), Conte, M. et al. *Nat. Commun.* **13**, 1–13 (2022)). In addition, we showed in previous studies that 3D trajectories generated with the model are compatible with real distance dynamics from live imaging experiments (Chiariello, A. M. et al., *Biophys. J.* **119**, 873–883 (2020)). Therefore, we are confident that model dynamics are a good proxy of real 3D trajectories and provide testable predictions for future experiments.

We stressed this point in the revised version, sections “*Time dynamics of 3D contacts is highly variable in SARS-CoV-2*”.

Reviewer: *The inclusion of the CTCF component in the model (Figure 6) presents complexity in understanding its relevance to Hi-C data, especially when the model only encompasses A and B nucleosomes. It remains unclear what insights Figure 6 provides or whether it is necessary for the overall understanding of the research.*

As clarified in the previous point, the model presented in Figure 6 does not include any A/B polymer beads but contains the binding sites obtained from HiC data in Mock and SARS-CoV-2 infected cells, for the DDX58 locus (analogously, Suppl. Fig. 6 about the IFIT locus). As specified, and now better clarified, in the section “*Structural re-arrangements of interferon response genes (IFN) loci*” and in the revised Methods section “*Polymer model of interferon response genes DDX58 and IFIT*”, those sites are inferred with a computational procedure named PRISMR (Polymer Recursive Inference Statistical Method, details in Bianco, S. et al. *Nat. Genet.* **50**, 662–667 (2018)) which minimizes a cost function that considers the difference between HiC matrix and a model contact matrix and a term that penalizes the presence of too many types of binding sites. In addition, loop-extrusion is integrated in the model with anchor points defined by CTCF ChIP-Seq (see Suppl. Fig. 4A and reference Conte, M. et al. *Nat. Commun.* **13**, 1–13 (2022)).

The analysis of binding sites and epigenetic marks shown in Figure 6 for DDX58 and Suppl. Fig. 6 for IFIT cluster highlight that architecture changes observed after SARS-CoV-2 infection not only involves CTCF and Cohesin (i.e. Loop extrusion) but also other factors, as histone marks, consistent with our results in which other mechanisms (i.e. phase-separation) are altered, beyond loop extrusion. Following Reviewer comment, we clarified better this in the revised manuscript, sections “*Chromatin re-arrangements in SARS-CoV-2 infection correlate with a combination of changes of CTCF and histone marks*” and “*Discussion*”.

Reviewer: *In summary, while the model can be adjusted to align with the observed weakening of TADs, its predictive power and utility remain unclear. It is uncertain whether a model that simply weakens all bindings could equally replicate the data for infected cells. Overall, the results may not be compelling enough to warrant publication in a high-profile journal.*

To strengthen the results of our work and to convince the reviewer about the validity of our study, we put a big effort in clarifying several aspects of the polymer model we employed, in particular about the A/B compartments. To highlight the power of this approach, we performed additional MD simulations, where we show that the model is able to explain the experimentally observed weakening of A compartment and increased A/B mixing in SARS-CoV-2 condition.

At the TADs level and IFN gene loci, we show that a model combining LE and PS mechanisms, which has been previously shown to be a good description of genome architecture (Conte, M. et al. *Nat. Commun.* **13**, 1–13 (2022)), predicts that SARS-CoV-2 infection alters both LE (extruder reduction) and PS (binding weakening). It is worth to note that it has been experimentally observed Cohesin reduction in SARS-CoV-2 infected cells (Wang, R. et al., *Nat. Microbiol.* **8**, 679–694 (2023)), in agreement with our model, as well as alteration of the protein interaction network triggered by the virus (Gordon, D. E. et al., *Nature* **583**, 459–468 (2020)), potentially affecting protein-chromatin interaction, as suggested by our study. This is an aspect never considered before that can be used to design novel experiments and represents the starting point for further investigations.

Overall, we hope that given all the clarifications above discussed and the improvements to the revised manuscript we strengthened our results and made our work suitable for publication.

Reviewer: Minor points:

1) Clarify the definition of “MOC” in line 80. It is unclear whether “mock” refers to cells infected by SARS-CoV-2 or cells before infection.

We thank the reviewer for the observation, we clarified this important aspect in the revised version.

2) Regarding DDX58 and IFIT genes, their expression during COVID-19 infection presents ambiguity. It is essential to determine whether the changes in the binding patterns of the DDX58 gene are caused by SARS genes or by the reorganization of cellular defense mechanisms. This distinction will help clarify whether the observed changes represent defects or gains, as suggested in the text around line 257.

This is an interesting point. As shown in Wang, R. et al., *Nat. Microbiol.* **8**, 679–694 (2023), SARS-CoV-2 infected cells tend to have a limited activation of IFN genes, which in general are expressed during an infection, as they act as response of the host cells to the pathogen. Furthermore, this is coupled with reduction of PolII at IFN genes, indicating that the alteration occurs at the level of transcription. Importantly, this is in agreement with immune-pathological features in patients with severe Covid-19 syndrome (Carvalho, T. et al. *Nat. Rev. Immunol.* **21**, 245–256 (2021)). HiC data of chromatin architecture after SARS-CoV-2 infection around those IFN genes help to understand their altered regulation and interpret it as an effect of viral chromatin re-structuring, rather than from cellular defense mechanisms. Our model of DDX58 and IFIT genes, applied on HiC data, supports this scenario suggesting that the changes in the binding patterns are likely caused by SARS-CoV-2 itself, through the alteration of Cohesin levels and protein binding, which in turn leads to a loss of coherence in the population of 3D structures and altered regulation.

In this regard, it is also interesting to note that HiC data (as well as our modelling approach applied on them) from cells infected with human common cold coronavirus HCoV-O43 (Wang, R. et al., *Nat. Microbiol.* **8**, 679–694 (2023)) reveal much more limited chromatin re-arrangements, suggesting again that chromatin structural changes are specifically induced by SARS-CoV-2 virus and not by defense mechanisms of the host cell.

We clarified this important aspect in the revised version, section “Single cell 3D structures result highly variable in SARS-CoV-2 infected condition”.

3) In the discussion, it might be more critical to specify whether the observed changes are due to viral genes attempting to directly reduce interferon (IFN) production or whether they result from cellular responses aiming to activate IFN genes.

In line with the previous point, our results point toward a scenario in which the virus strategically changes the chromatin architecture to inhibit IFN genes activity which in turn weakens the host cell immunological response. This would explain why in severe acute Covid-19 patients the IFN response to the inflammation is weak and delayed (Carvalho, T. et al. *Nat. Rev. Immunol.* **21**, 245–256 (2021)). Indeed, other viruses, such as human common cold coronavirus HCoV-O43 do not exhibit the same changes. As suggested by the reviewer, we better stressed this point in the “Discussion”.

Reviewer #3

Reviewer: This paper sought to address a significant issue: the impact of SARS-CoV-2 infection on the host cell's chromatin architecture. The authors effectively highlight gaps in our understanding of the biophysical mechanisms by which the virus infection alters 3D chromatin structure. The study aims to fill this gap by employing polymer physics models and Molecular Dynamics (MD) simulations to study chromatin reorganizations at multiple genomic scales. The paper is ambitious in its scope, aiming to elucidate the mechanistic underpinnings of how SARS-

CoV-2 affects chromatin organization and, by extension, gene regulation. It offers insights into the architecture of specific genomic loci linked to the antiviral interferon response, which is of high immunological interest. Some findings by modeling are very interesting and cannot be readily obtained with current experimental methods or datasets, for example, the looping dynamics between genes and enhancers and the three-way interactions. Overall, this study is well-executed, clearly-written, and sets the stage for a potentially impactful study that could provide valuable insights into the epigenomic effects of SARS-CoV-2 infection.

We thank the reviewer for her/his very positive opinion about our work, considering “valuable insights”. Below we report our reply to her/his specific comments.

Reviewer: However, I have some comments that the authors shall consider addressing before publication:

1). In the first A/B compartment modeling section, the authors employ a simplified block co-polymer model. While this is a starting point, it may not adequately reflect the biophysical properties of a real genome. I suggest using a polymer model that mimics an actual chromosome (or part of it) with real A/B patterns to better support the conclusions.

We thank the reviewer for this comment. Following her/his suggestion, we generated polymer models for the entire chromosome 11 (Figure R3, we report again the figure here for sake of clarity). To this aim, we classified A/B beads using the 1st eigenvector from PCA analysis of HiC data from the Mock condition (100kb resolution, Figure R3A). Then, we performed additional Molecular Dynamics simulations with relative affinities E_{A-A}/E_{A-B} in the range 1.06-1.08 (E_{A-A} in the range 9.5-9.7 $K_B T$, i.e., as the model in Figure 1) and considered balanced ($E_{B-B} > E_{A-A}$) and unbalanced ($E_{B-B} = E_{A-A}$) affinities. We then obtained the best combination of balanced and unbalanced models fitting the experimental saddle-plots, where, consistently with Figure 1E, SARS-CoV-2 data are better described by a combination with 80% of $E_{B-B} > E_{A-A}$ (Figure R4A) against 30% in Mock case (see next point for more detail). The contact pattern is well reproduced as well as the compartment profile (Pearson correlation $r=0.9$ in both Mock and SARS-CoV-2 infected condition). We included and discussed these new results in revised manuscript, section “Modelling of chromatin re-structuring in A/B compartments”, in Methods section “Polymer model of A/B compartments” and included those results in the new Extended Fig. 2.

Figure R3: Polymer model including A/B compartment profile recapitulates genome organization in Mock and SARS-CoV-2 conditions. **A:** We defined a polymer model based on 1st eigenvector, from which A/B compartments are defined. **B:** Contact maps simulated in Mock condition accurately recapitulates experimental HiC data (Pearson $r=0.94$, distance corrected $r'=0.5$). In the middle, 1st eigenvector of the PCA analysis from HiC (up) and model (bottom) contact maps (Pearson $r=0.89$). **C:** As panel B, for SARS-CoV-2 infected condition (Pearson $r=0.9$, distance corrected $r'=0.5$, between HiC and model contact maps, Pearson $r=0.89$ between experimental and model 1st eigenvectors).

2). The authors state that balanced A-A/B-B interactions dominate in the mock genome. This seems to contradict the prevailing notion that homotypic interactions and mediating machinery between A-A and B-B compartments are distinct. This point needs justification.

We thank the reviewer for this observation. For sake of simplicity, to study genome-wide compartmentalization changes, we modelled A/B compartments as a simple block-copolymer with homotypic affinities in the range $9.3-9.7K_B T$. In this case, we performed a fit to optimize the genome-wide saddle-plot with a combination of models with unbalanced ($E_{B-B} > E_{A-A}$) and balanced ($E_{B-B} = E_{A-A}$) affinities and found that Mock is best described by a combination with higher component of balanced $E_{B-B} = E_{A-A}$ (almost 90%) affinities. It is important to observe that this analysis does not consider local complexities where distinction between A and B compartment could emerge more strongly and reveal the different nature of affinities driving their formation.

To investigate this aspect, we performed the same analysis using as real case of study chromosome 11 (see previous point), and found that Mock saddle-plot of chromosome 11 is well fitted by a combination with 70% of balanced $E_{A-A} = E_{B-B}$ and 30% unbalanced $E_{B-B} > E_{A-A}$ models (Figure R4A, B), in line with the genome-wide result but also highlighting a not negligible role for unbalanced interactions even in the Mock case, consistent with the notion of distinct machineries for A and B compartments. To test the robustness of the result, we verified that the same combination is found by optimizing the 1st eigenvector profile.

We stressed better this point in the revised manuscript, section “Modelling of chromatin restructuring in A/B compartments” and in Methods section “Polymer model of A/B compartments”.

Figure R4: Combination of balance and unbalanced models explains in Mock and SARS-CoV-2 conditions. **A:** Best fit combination of models with balanced and unbalanced homotypic affinities (E_{A-A}/E_{A-B} in the range 1.06-1.08, $E_{B-B}/E_{A-B}=1.08$) in Mock and SARS-CoV-2 infected condition. **B:** Saddle-plots from HiC data and model of chromosome 11 in Mock (Pearson correlation $r=0.92$) and SARS-CoV-2 infected condition (Pearson correlation $r=0.90$).

3). Similar to my first comment, the loop extrusion modeling in Fig. 2 appears to use an artificial locus. Given that loop extrusion and phase separation may vary between euchromatin and heterochromatin, it would be beneficial to model these processes using actual chromatin regions.

Thanks for this observation. The aim of this analysis in Figure 2 is to quantitatively understand the changes exhibited upon SARS-CoV-2 infection by the genome-wide average of contact probability decay. To this aim, as correctly pointed out by the reviewer, we used a simple toy model with an array of consecutive, equally sized TADs, without taking into account A/B compartments. Therefore, following her/his suggestion, we performed new simulations of a toy model including A/B compartments and TADs (Figure R5, we report again the figure here for sake of clarity). Specifically, we studied the compartmentalization strength in different conditions. As reference level for the model parameters, we employed affinities previously used (i.e. sampled in the range $9-9.7K_B T$, Figure 1) and a number of extruding factors compatible with previous studies (Fudenberg, G. et al., *Cell Rep.* **15**, 2038–2049 (2016)), i.e. LEFs average separation approximately 200kb, and in line with the results discussed in Figure 2, which represent our control condition (Figure R1, panel B, left matrix). When the number of extruders is dramatically lowered (about 4-fold reduction), the affinities kept fixed, the saddle-plot analysis reveals strengthened compartmentalization (Figure R2, panel B, middle matrix) along with intraTAD contact weakening ($p\text{-val}=10^{-46}$, one-sided Mann-Whitney U test), in full agreement with the experimental observation

by which deletion of Cohesin increases compartment strength (Schwarzer, W. et al., *Nature* **551**, 51–56 (2017), Rao, S. S. P. et al., *Cell* **171**, 305-320.e24 (2017)). Interestingly, if the same decrease of extruders is coupled with a decrease of the homo-typic affinity A-A and B-B (we acted on both for sake of simplicity), both intraTAD contacts ($p\text{-val}=10^{-97}$, one-sided Mann-Whitney U test) and compartmentalization level are reduced (Figure R2, panel B, right matrix), fully consistent with HiC data from SARS-CoV-2 infected cells. Those results are quantitatively highlighted by the compartmentalization strength (*bioRxiv* 2022.10.31.514564 (2022)) profile in the three conditions (Figure R2, panel C).

We agree that a polymer model like this for an actual chromatin region would be very interesting to simulate, but it would require high resolution data to simulate a reliable loop extrusion on a very large region to include A/B compartments, going beyond the purpose of the current paper.

We better stressed this point in the revised manuscript, section “*Viral infection impacts loop-extrusion and phase-separation features at TAD level*”.

Figure R5. Combined alteration of LE and PS explains compartment weakening in SARS-CoV-2 infected cells. **A:** Polymer model including either LE, driving TAD formation, and PS driving A/B compartment formation. **B:** Saddle-plots corresponding to different system parameters (distance between extruders or, equivalently their number, and homotypic (E_{A-A} and E_{B-B}) affinity) results in altering compartmentalization level. Reduction of extruders only leads to compartment strengthening. Combined reduction of extruders and affinity leads to compartment weakening. **C:** Compartment strength profile for the different model parameters.

4). *The part of three-way interactions is intriguing. However, it would strengthen the paper if the authors could demonstrate that the frequencies of these interactions are statistically higher than random occurrences based on their modeling data.*

We thank the reviewer for this observation. As we generate populations of single-molecule structures from MD simulations, we could consider the distributions of three-way contacts in Mock and SARS-CoV-2 infected models. Therefore, following Reviewer’s suggestion, we focused on the specific triple contact involving E1-IFIT3-E3 in Mock model (indicated with the arrow in Figure 5E). Then, to show that the frequencies of those interactions are statistically significant, we considered as control level another triplet having the same genomic distances but the central point of view taken at the boundary between two domains downstream the IFIT3 promoter. We find that the distributions result significantly different (Mann-Whitney U test $p\text{-val}=3*10^{-9}$), other control triplets returned similar results.

Analogously, we find that the specific triple contact involving E1-IFIT3-E3 in Mock exhibited statistically higher interactions with respect to SARS-CoV-2 distribution (Mann-Whitney U $p\text{-val}=7*10^{-5}$).

We reported this in the revised manuscript, section “*Single cell 3D structures result highly variable in SARS-CoV-2 infected condition*” and in the Methods section “*Analysis of contact maps, saddle-plots and contact probability*”.

5). *In the section on enhancer-promoter time dynamics, the authors note that contact time is generally shorter in infected cells. Could the authors also discuss the changes of contact frequencies (intervals between each contact)? And which (contact times or frequencies) contributed more to the changes on bulk contact maps?*

We thank the reviewer for this point. As suggested, we re-analyzed our time trajectories focusing on the distribution of intervals between consecutive contacts (Figure R6, panel A) and we find that, consistently with results shown in Figure 5, panel F, the trajectories from the polymer model in Mock cells exhibit lower time intervals ($p\text{-val}=3 \times 10^{-4}$, one-sided t-test) between each contact than SARS-CoV-2 infected condition. The distributions of frequencies and contact times are very similar with a 1.6-fold increase of the average time interval, suggesting that they similarly contribute to the change in the bulk contact maps in the infected case.

We added this new analysis in revised manuscript, section “*Time dynamics of 3D contacts is highly variable in SARS-CoV-2*” and added a new panel E in Suppl. Fig. 5.

Figure R6: Distribution of time intervals between two contacts in Mock and SARS-CoV-2 infected conditions. We analyze time intervals τ between two contacts between DDX58 gene and its enhancer (left panel). Distributions of time intervals are lower in Mock with respect to SARS-CoV-2 condition ($p\text{-val}=3 \times 10^{-4}$, one-sided t-test).

6). *The binding site analyses in Figure 6 shows graphical minor changes across four types of binding sites. Statistical testing could help clarify the significance of these differences. Additionally, the criteria for determining the number of types of binding sites should be clarified. Would further classification of binding sites show clearer patterns?*

Thanks for this suggestion. We performed statistical tests to quantify the differences between binding sites in Mock and SARS-CoV-2 conditions. To give insights about the biological nature of those changes, we focused on their relationship with epigenetic features. Specifically, we generated a random control model with random permutation of binding sites and performed the same analysis. To ensure robustness of the analysis, 1000 random permutations were considered. In this way, we could identify the most significant changes in association between type and epigenetic mark. We find that main changes involve binding sites #1, #2 and #4 correlating with CTCF ($p\text{-val}=0.047$), Cohesin ($p\text{-val}=0.065$) and H3K4ac ($p\text{-val}=0.033$).

Significance analysis has been added in the revised manuscript, section “*Chromatin rearrangements in SARS-CoV-2 infection correlate with a combination of changes of CTCF and histone marks*”, Methods section “*Epigenetics signature of binding sites*” and Figure 6.

The criteria for determining the number of types on the polymer relies on a computational procedure named PRISMR (Polymer Recursive Inference Statistical Method, details Bianco et al 2018) which minimizes a cost function that considers the difference between HiC matrix and a predicted matrix and a term that penalizes the presence of too many types. Therefore, addition of more types would not lead to clearer patterns. We included more information about the procedure to determine number and location of binding sites in the revised manuscript, Method section “*Polymer model of interferon response genes DDX58 and IFIT*”.

7). *For the benefit of a broader audience, I recommend that the authors provide more detailed explanations of their analyses in the methods section, particularly concerning the “Epigenetic signature of binding sites” and three-way interactions.*

We appreciate Reviewer’s recommendation and we accordingly expanded the revised Methods, with more details about the three-way contacts in the section “*Analysis of contact maps, saddle-plots and contact probability*” and about the cross-correlation analysis of the binding sites with epigenetics features, section “*Epigenetics signature of binding sites*”.

8). *Providing a step-by-step notebook for readers to better understand and reproduce the modeling and analyses would be a very valuable addition.*

Codes to run the MD simulations for A/B compartments, TADs and DDX58 locus have been provided as Supplementary files with a README included. Following Reviewer suggestion, we provided in a Notebook all the necessary step-by-step information to reproduce analysis of simulated polymer structures and 3D trajectories discussed in the paper.

REVIEWERS' COMMENTS

Reviewer #1 (Remarks to the Author):

The authors addressed my concerns in a satisfactory way and made a compelling argument that figuring out why SARS-CoV-2 infection induces the observed changes in genome architecture is beyond the scope of this manuscript. Thus, I am in favor of proceeding with publication.

Reviewer #2 (Remarks to the Author):

The new version of the manuscript show that A/B modeling could be a useful tool to discuss the deep cellular response to virus infection, and perhaps help us to differentiate mechanisms.

Overall I think the revised version in now a useful contribution to this very complicated research area, and I recommend publication in Nature communication.

Reviewer #3 (Remarks to the Author):

The revised manuscript has been significantly improved and addressed my previous questions.

Response to Reviewers

We thank the Reviewers for their positive opinion about the revised manuscript and for their support to publication. We would also like to thank them for all their useful suggestions, giving us the opportunity to improve our work.